# Support for a clade of Placozoa and Cnidaria in genes with minimal compositional bias

Christopher E Laumer[1,2]*, Harald Gruber-Vodicka[3], Michael G Hadfield[4], Vicki B Pearse[5], Ana Riesgo[6], John C Marioni[1,2,7], Gonzalo Giribet[8]

[1]Wellcome Trust Sanger Institute, Hinxton, United Kingdom; [2]European Molecular Biology Laboratories-European Bioinformatics Institute, Hinxton, United Kingdom; [3]Max Planck Institute for Marine Microbiology, Bremen, Germany; [4]Kewalo Marine Laboratory, Pacific Biosciences Research Center and the University of Hawaii-Manoa, Honolulu, United States; [5]Institute of Marine Sciences, University of California, Santa Cruz, United States; [6]Invertebrate Division, Life Sciences Department, The Natural History Museum, London, United Kingdom; [7]Cancer Research UK Cambridge Institute, University of Cambridge, Cambridge, United Kingdom; [8]Museum of Comparative Zoology, Department of Organismic and Evolutionary Biology, Harvard University, Cambridge, United States

**Abstract** The phylogenetic placement of the morphologically simple placozoans is crucial to understanding the evolution of complex animal traits. Here, we examine the influence of adding new genomes from placozoans to a large dataset designed to study the deepest splits in the animal phylogeny. Using site-heterogeneous substitution models, we show that it is possible to obtain strong support, in both amino acid and reduced-alphabet matrices, for either a sister-group relationship between Cnidaria and Placozoa, or for Cnidaria and Bilateria as seen in most published work to date, depending on the orthologues selected to construct the matrix. We demonstrate that a majority of genes show evidence of compositional heterogeneity, and that support for the Cnidaria + Bilateria clade can be assigned to this source of systematic error. In interpreting these results, we caution against a peremptory reading of placozoans as secondarily reduced forms of little relevance to broader discussions of early animal evolution.
DOI: https://doi.org/10.7554/eLife.36278.001

*For correspondence: claumer@ebi.ac.uk

Competing interests: The authors declare that no competing interests exist.

## Introduction

The discovery (*Schulze, 1883*) and mid-20th century rediscovery (*Grell and Benwitz, 1971*) of the enigmatic, amoeba-like placozoan *Trichoplax adhaerens* did much to ignite the imagination of zoologists interested in early animal evolution (*Bütschli, 1884*). As microscopic animals adapted to extracellular grazing on the biofilms over which they creep (*Wenderoth, 1986*), placozoans have a simple anatomy suited to exploit passive diffusion for many physiological needs, with only six morphological cell types discernible even to intensive microscopical scrutiny (*Grell and Ruthmann, 1991*; *Smith et al., 2014*), albeit a greater diversity of cell types is apparent through single-cell RNA-seq (*Sebé-Pedrós, 2018a*). They have no conventional muscular, digestive, or nervous systems, yet show tightly-coordinated behaviour regulated by peptidergic signaling (*Smith et al., 2015*; *Senatore et al., 2017*; *Varoqueaux, 2018*; *Armon et al., 2018*). In laboratory conditions, they proliferate through fission and somatic growth. Evidence for sexual reproduction remains elusive, despite genetic evidence of recombination (*Srivastava et al., 2008*) and descriptions of early

**eLife digest** Filter-feeding sponges and tiny gliding, pancake-like animals called placozoans are the only two major groups of animals that lack muscles, nerves and an internal gut. Sponges have historically been seen as the first to have branched off in animal phylogeny – the family tree of living organisms that shows how species are related. This is because it is assumed that they split from the other animals before features including muscles, nerves and internal guts evolved.

Sequences of their genetic material (the genome) support this view, although some argue that jellyfish-like animals called ctenophores branched first. One explanation for this disagreement is that ctenophores use different proportions of amino acids in their proteins, known as compositional heterogeneity. Computer algorithms that assume amino acid usage is the same universally throughout evolution may therefore place ctenophores incorrectly. In contrast, so far the only genome from a placozoan shows that they are equally closely related to jellyfish and corals (cnidarians) and bilaterians, which includes worms, insects and vertebrates.

To test whether this view of the first branches of the animal tree of life is correct, Laumer et al. included the genomes from several undescribed species of placozoans in a phylogenetic analysis. These analyses showed a relationship that had not previously been seen. The placozoans were the closest living relative to cnidarians. However, when looking at the level of genes rather than whole genomes, the more usual relationship of placozoans being equally related to cnidarians and bilaterians re-emerged. To resolve this conflict, Laumer et al. focused on the genes that had the least compositional heterogeneity. When doing this, the relationship appeared to be the newly identified one of placozoans being most closely related to cnidarians.

Researchers studying cnidarians often hope to find some clues as to how the complex features they seem to share with bilaterians originated. The findings of Laumer et al. may suggest that the ancestors of the placozoans did in fact have muscles, nerves and guts, but they lost these traits in favor of a simpler lifestyle. An alternative, but controversial possibility is that the ancestor of cnidarians and bilaterians was a simple organism like a placozoan, and the two evolved their complex traits independently. The findings show a complex picture of early animal evolution. Further study of placozoans may well clarify this picture.

DOI: https://doi.org/10.7554/eLife.36278.002

abortive embryogenesis (*Eitel et al., 2011*; *Grell, 1972*), with the possibility that sexual phases of the life cycle may occur only under poorly understood field conditions (*Pearse and Voigt, 2007*; *McFall-Ngai et al., 2013*)

Given their simple, puzzling morphology and dearth of embryological clues, molecular data are crucial in placing placozoans phylogenetically. The position of Placozoa in the animal tree proved recalcitrant to early standard-marker analyses (*Kim et al., 1999*; *Silva et al., 2007*; *Wallberg et al., 2004*), although this paradigm did reveal a large degree of molecular diversity in placozoan isolates from around the globe, clearly indicating the existence of many cryptic species (*Pearse and Voigt, 2007*; *Eitel et al., 2013*; *Signorovitch et al., 2007*) with up to 27% genetic distance in *16S rRNA* alignments (*Eitel and Schierwater, 2010*). An apparent answer to the question of placozoan affinities was provided by analysis of a nuclear genome assembly (*Srivastava et al., 2008*), which strongly supported a position as the sister group of a clade of Cnidaria + Bilateria (sometimes called Planulozoa). However, this effort also revealed a surprisingly bilaterian-like (*Dunn et al., 2015*) developmental gene toolkit in placozoans, a paradox for such a simple animal.

As metazoan phylogenetics has pressed onward into the genomic era, perhaps the largest controversy has been the debate over the identity of the sister group to the remaining metazoans, traditionally thought to be Porifera, but considered to be Ctenophora by Dunn et al (*Dunn et al., 2008*). and subsequently by additional studies (*Hejnol et al., 2009*; *Moroz et al., 2014*; *NISC Comparative Sequencing Program et al., 2013*; *Whelan et al., 2015*; *Whelan et al., 2017*). Others have suggested that this result arises from artifacts with potentially additive effects, such as inadequate taxon sampling, flawed matrix husbandry (undetected paralogy or contamination), and use of poorly fitting substitution models (*Philippe et al., 2009*; *Pick et al., 2010*; *Pisani et al., 2015*; *Simion et al., 2017*; *Feuda et al., 2017*). A third view has emphasized that using different sets of

genes can lead to different conclusions, with only a small number sometimes sufficient to drive one result or another (*Nosenko et al., 2013*; *Shen et al., 2017*). This controversy, regardless of its eventual resolution, has spurred serious contemplation of possibly independent origins of several hallmark traits such as striated muscles, digestive systems, and nervous systems (*Moroz et al., 2014*; *Dayraud et al., 2012*; *Hejnol and Martín-Durán, 2015*; *Liebeskind et al., 2017*; *Moroz and Kohn, 2016*; *Presnell et al., 2016*; *Steinmetz et al., 2012*).

Driven by this controversy, new genomic and transcriptomic data from sponges, ctenophores, and metazoan outgroups have accrued, while new sequences and analyses focusing on the position of Placozoa have been slow to emerge. Here, we provide a novel test of the phylogenetic position of placozoans, adding draft genomes from three putative species that span the root of this clade's known diversity (*Eitel et al., 2013*) (*Table 1*), and critically assessing the role of systematic error in placing of these enigmatic organisms (*Laumer, 2018*).

## Results and discussion

Orthology assignment on sets of predicted proteomes derived from 59 genome and transcriptome assemblies yielded 4294 gene trees with at least 20 sequences each, sampling all five major metazoan clades and outgroups, from which we obtained 1388 well-aligned orthologues. Within this set, individual maximum-likelihood (ML) gene trees were constructed, and a set of 430 most-informative orthologues were selected on the basis of tree-likeness scores (*Misof et al., 2013*). This yielded an amino-acid matrix of 73,547 residues with 37.55% gaps or missing data, with an average of 371.92 and 332.75 orthologues represented for Cnidaria and Placozoa, respectively (with a maximum of 383 orthologues present for the newly sequenced placozoan H4 clade representative; *Figure 1*).

Our Bayesian analyses of this matrix place Cnidaria and Placozoa as sister groups with full posterior probability under the general site-heterogeneous CAT + GTR + Γ4 model (*Figure 1*). Under ML inference with the C60 +LG + FO + R4 profile mixture model (*Wang et al., 2018*) (*Figure 1—figure supplement 1*), we again recover Cnidaria + Placozoa, albeit with more marginal resampling support. Both Bayesian and ML analyses show little internal branch diversity within Placozoa. Accordingly, deleting all newly-added placozoan genomes from our analysis has no effect on topology and only a marginal effect on support in ML analysis (*Figure 1—figure supplement 2*). Quartet-based concordance analyses (*Zhou, 2017*) show no evidence of strong phylogenetic conflicts among ML gene trees in this 430-gene set (*Figure 1*), although internode certainty metrics are close to 0 for many key clades including Cnidaria + Placozoa, indicating that support for some ancient relationships may be masked by gene-tree estimation errors, emerging only in combined analysis (*Gatesy and Baker, 2005*).

Compositional heterogeneity of amino-acid frequencies along the tree is a source of phylogenetic error not modelled by even complex site-heterogeneous substitution models such as CAT+GTR

**Table 1.** Summary statistics describing the contiguity and completeness of the draft host metagenome bins from the three clade A placozoan isolates utilized in this paper, presented in comparison to the reference H1 strain.

| | H11 | H4 | H6 | H1 |
|---|---|---|---|---|
| assembly span (Mbp) | 56.63 | 83.39 | 76.7 | 98.06 |
| scaffold number | 5813 | 5337 | 8310 | 1415 |
| scaffold N50 (kbp) | 12.738 | 25.97 | 12.84 | 5790 |
| GC% | 30.76 | 30.84 | 29.9 | 29.37 |
| BUSCO2 Eukaryota complete (of 303) | 220 | 276 | 239 | 294 |
| BUSCO2 Eukaryota complete + partial (of 303) | 246 | 282 | 265 | 298 |
| Average # of hits per BUSCO | 1.00 | 1.04 | 1.00 | 1.00 |
| % of BUSCOs with more than one match | 0.45 | 3.99 | 0.42 | 0.34 |

DOI: https://doi.org/10.7554/eLife.36278.003

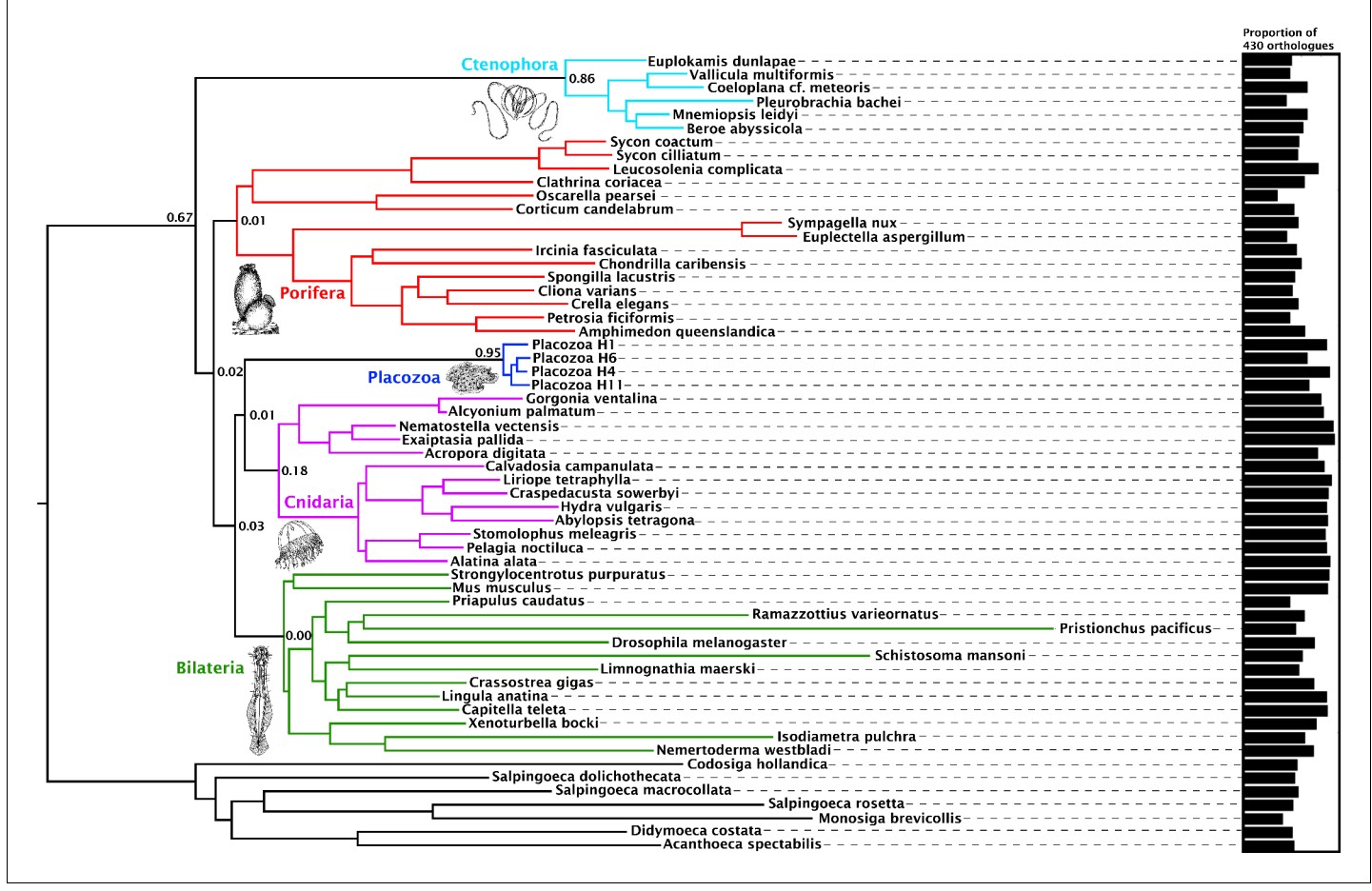

**Figure 1.** Consensus phylogram showing deep metazoan interrelationships under Bayesian phylogenetic inference of the 430-orthologue amino acid matrix, using the CAT + GTR + Γ4 mixture model. All nodes received full posterior probability. Numerical annotations of given nodes represent Extended Quadripartition Internode Certainty (EQP-IC) scores, describing among-gene-tree agreement for both the monophyly of the five major metazoan clades and the given relationships between them in this reference tree. A bar chart on the right depicts the proportion of the total orthologue set each terminal taxon is represented by in the concatenated matrix. 'Placozoa H1' in this and all other figures refers to the GRELL isolate sequenced in *Srivastava et al., 2008*, which has there and elsewhere been referred to as *Trichoplax adhaerens*, despite the absence of type material linking this name to any modern isolate. Line drawings of clade representatives are taken from the BIODIDAC database (http://biodidac.bio.uottawa.ca/).

DOI: https://doi.org/10.7554/eLife.36278.004

The following figure supplements are available for figure 1:

**Figure supplement 1.** Maximum likelihood tree under the C60 +LG + FO + R4 profile mixture model, inferred from the 430-orthologue matrix with full taxon sampling.

DOI: https://doi.org/10.7554/eLife.36278.005

**Figure supplement 2.** Maximum likelihood tree under a profile mixture model inferred from the 430-orthologue matrix, with only Placozoa H1 used to represent this clade.

DOI: https://doi.org/10.7554/eLife.36278.006

(*Blanquart and Lartillot, 2008*; *Foster, 2004*; *Lartillot and Philippe, 2004*; *Lartillot et al., 2013*). Furthermore, previous analyses (*Nosenko et al., 2013*) have shown that placozoans and choanoflagellates in particular, both of which taxa our matrix samples intensively, deviate strongly from the mean amino-acid composition of Metazoa, perhaps as a result of genomic GC content discrepancies. As a measure to at least partially ameliorate such nonstationary substitution, we recoded the amino-acid matrix into the 6 'Dayhoff' categories, a common strategy previously shown to reduce the effect of compositional variation among taxa, albeit the Dayhoff-6 groups represent only one of many plausible recoding strategies, all of which sacrifice information (*Feuda et al., 2017*; *Nesnidal et al., 2010*; *Rota-Stabelli et al., 2013*; *Susko and Roger, 2007*). Analysis of this recoded matrix under

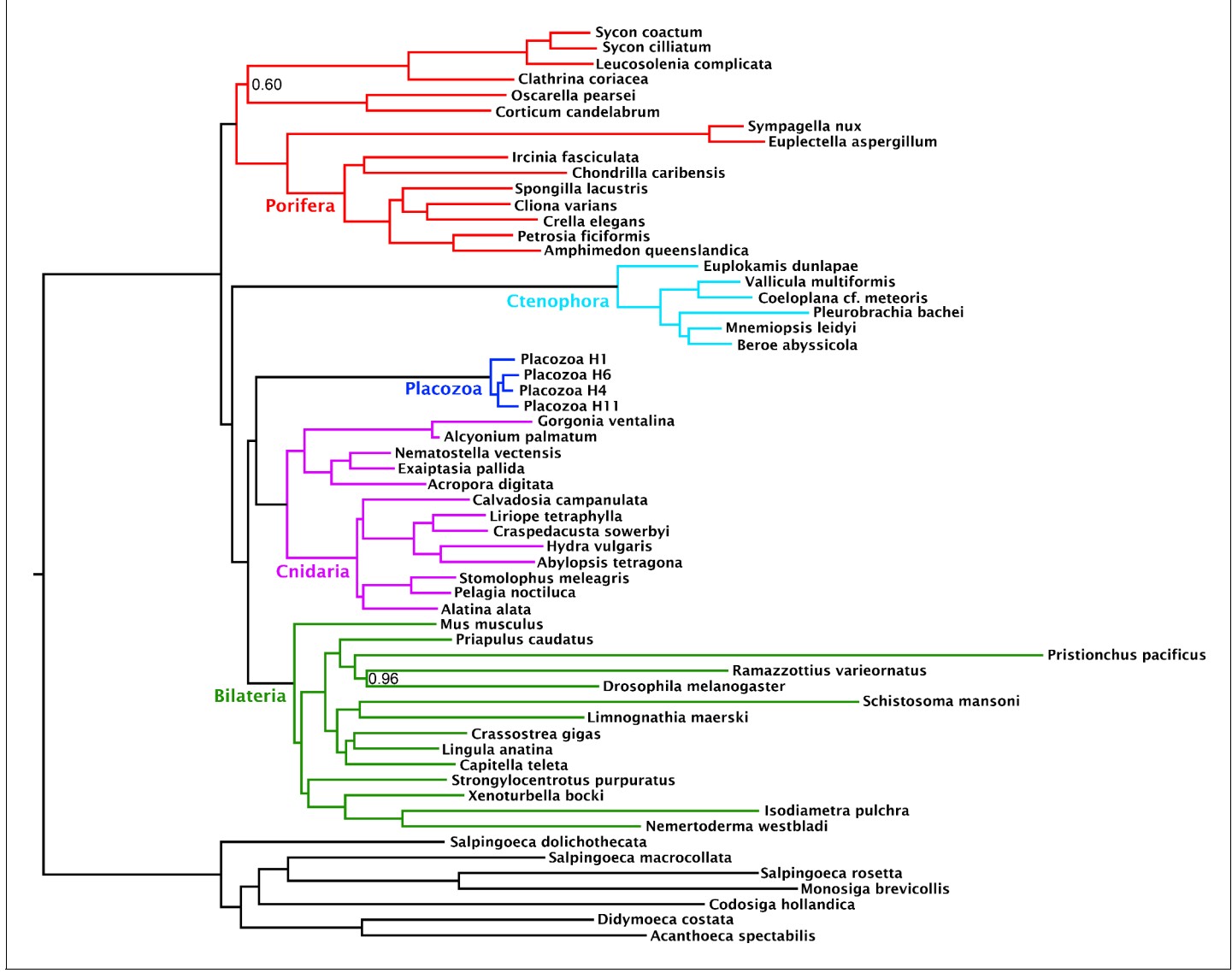

**Figure 2.** Consensus phylogram under Bayesian phylogenetic inference under the CAT + GTR + Γ4 mixture model, on the 430-orthologue concatenated amino acid matrix, recoded into 6 Dayhoff groups. Nodes annotated with posterior probability; unannotated nodes received full support.

DOI: https://doi.org/10.7554/eLife.36278.007

the CAT + GTR model again recovered full support (pp = 1) for Cnidaria + Placozoa (*Figure 2*). Indeed, under Dayhoff-6 recoding, the only major change is in the relative positions of Ctenophora and Porifera, with the latter here constituting the sister group to all other animals with full support. Similar recoding-driven effects on relative positions of Porifera and Ctenophora have also been seen in other recent work (*Feuda et al., 2017*), and have been interpreted to indicate a role for compositional bias in misplacing Ctenophora as sister group to all other animals

Many research groups, using good taxon sampling and genome-scale datasets, and even recently including data from a new divergent placozoan species (*Whelan et al., 2017*; *Feuda et al., 2017*; *Eitel, 2017*), have consistently reported strong support for Planulozoa under the CAT + GTR model. Indeed, when we construct a supermatrix from our predicted peptide catalogues using a different strategy, relying on complete sequences of 303 pan-eukaryote 'Benchmarking Universal Single-Copy Orthologs' (BUSCOs) (*Simão et al., 2015*), we also see full support in a CAT + GTR + Γ analysis for Planulozoa, in both amino-acid (*Figure 3a*) and Dayhoff-6 recoded alphabets (*Figure 3b*). Which

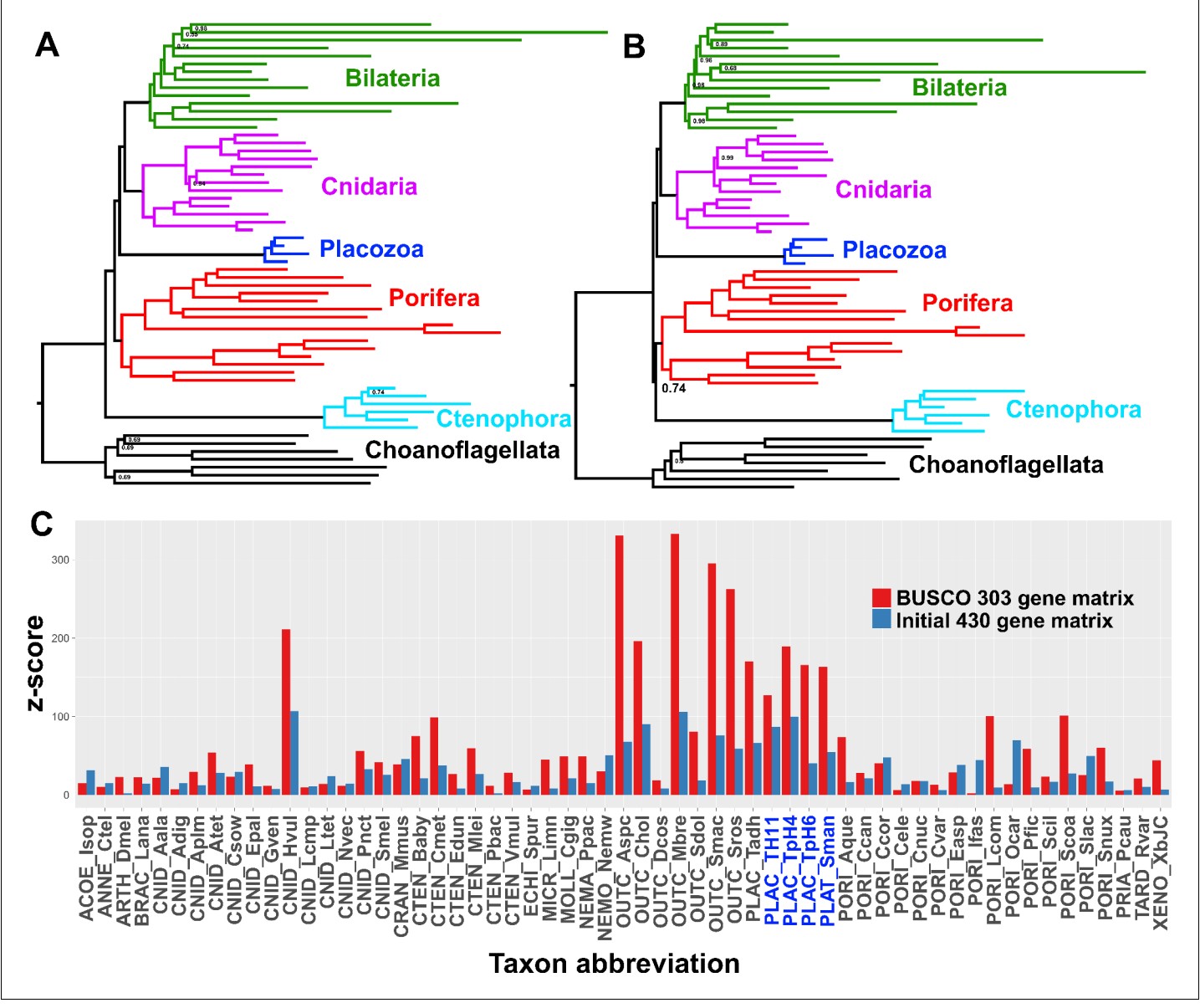

**Figure 3.** Posterior consensus trees from CAT + GTR + Γ4 mixture model analysis of a 94,444 amino acid supermatrix derived from the 303 single-copy conserved eukaryotic BUSCO orthologs, analysed in A.  amino acid space or (**B**) the Dayhoff-6 reduced alphabet space. Nodal support values comprise posterior probabilities; nodes with full support not annotated. Taxon colourings as in previous Figures. (**C**) Plot of z-scores (summed absolute distance between taxon-specific and global empirical frequencies) from representative posterior predictive tests of amino acid compositional bias, from both the BUSCO 303-orthologue matrix (red) and the initial 430-orthologue matrix (blue). Placozoan taxon abbreviations are shown in blue font.

DOI: https://doi.org/10.7554/eLife.36278.008

phylogeny is correct, and what process drives support for the incorrect topology? Posterior predictive tests, which compare the observed among-taxon usage of amino-acid frequencies to expected distributions simulated using the sampled posterior distribution and a single composition vector, may provide insight (*Feuda et al., 2017*; *Lartillot and Philippe, 2004*). Both the initial 430-gene matrix and the 303-gene BUSCO matrix fail these tests, but the BUSCO matrix fails it more profoundly, with z-scores (measuring mean-squared across-taxon heterogeneity) scoring in the range of 330–340, in contrast to the range of 176–187 seen in the 430-gene matrix (*Table 2*). Furthermore, inspecting z-scores for individual taxa in representative chains from both matrices shows that a large

**Table 2.** Mean (and standard deviation of) z-scores from posterior predictive tests of per-site amino acid diversity and among-lineage compositional homogeneity, called for amino-acid alignments using the PhyloBayes-MPI v1.8 readpb_mpi –div and –comp options, respectively, with burn-ins selected as per the posterior consensus summaries shown elsewhere.

Except for the diversity statistic in the test-passing matrix, all tests reject (at p=0.05) the adequacy of the inferred CAT + GTR + Γ4 model to describe the data.

| | Diversity | Composition (mean) | Composition (maximum) |
|---|---|---|---|
| 430 matrix | 1.94 (0.09) | 181.35 (7.50) | 105.04 (3.13) |
| BUSCO 303-gene matrix | 11.27 (0.73) | 334.98 (4.56) | 107.56 (6.17) |
| comp-failed matrix | 2.51 (0.19) | 270.16 (12.03) | 173.87 (9.15) |
| comp-passed matrix | 0.81 (0.18) | 107.67 (10.10) | 63.19 (6.95) |

DOI: https://doi.org/10.7554/eLife.36278.009

amount of this global difference in z-scores can be attributed to placozoans, with additional contributions from choanoflagellates and select isolated representatives of other clades (*Figure 3C*).

As a final measure to describe the influence of compositional heterogeneity in this dataset, we applied a null-simulation test for compositional bias to each alignment in our set of 1388 orthologues. This test, which compares the real data to a null distribution of amino-acid frequencies simulated along assumed gene trees with a substitution model using a single composition vector, is less prone to Type II errors than the more conventional X (*Grell and Benwitz, 1971*) test (*Foster, 2004*). Remarkably, at a conservative significance threshold of α = 0.10, the majority (764 genes or ~55%) of this gene set is identified as compositionally biased by this test, highlighting the importance of using appropriate statistical tests to control this source of systematic error, rather than applying arbitrary heuristic cutoffs (*Kück and Struck, 2014*). Building informative matrices from gene sets on either side of this significance threshold, and again applying both CAT + GTR mixture models and ML profile mixtures, we see strong support for Cnidaria + Placozoa in the test-passing supermatrix, and conversely, strong support for Cnidaria + Bilateria in the test-failing supermatrix (*Figure 4*, *Figure 4—figure supplement 1*, *Figure 4—figure supplement 2*). Interestingly, in trees built through CAT + GTR + Γ4 analysis of the test-failing supermatrix (*Figure 4A,C*), in both amino-acid and Dayhoff-6 alphabets, we also observe full support for Porifera as sister to all other animals. In contrast, analysis of this amino acid matrix under a profile mixture model recovers support for Ctenophora in this position (*Figure 4—figure supplement 1*), indicating that, at least for this alignment, compositional heterogeneity need not be invoked to explain why outcomes differ among analyses, as some have argued (*Feuda et al., 2017*): both CAT + GTR and the C60 +LG + FO + R4 profile mixture model assume a single composition vector over time, but the CAT + GTR model is better able to accommodate site-heterogeneous substitution patterns (*Lartillot et al., 2013*; *Quang et al., 2008*). In the context of this experiment, Dayhoff-6 recoding appears impactful only for the test-passing supermatrix (*Figure 4B,D*), where it obviates support for Ctenophora-sister (*Figure 4B*, *Figure 4—figure supplement 2*) in favour of (albeit, with marginal support) Porifera-sister (*Figure 4D*), and also diminishes support for Placozoa + Cnidaria (in contrast to the 430-gene matrix; *Figure 2*), perhaps reflecting the inherent information loss of using a reduced amino-acid alphabet for this relatively shorter matrix.

A possible hidden variable related to the phylogenetic discordance we describe, the precise significance of which remains unclear, is mean trimmed alignment length: both the test-passing and the original 430-gene matrix are composed of considerably shorter alignments than the test-failing and the 303-gene BUSCO matrix (see Materials and methods). Indeed, alignment length has been previously shown to be predictive of a number of other metrics of phylogenetic relevance (*Shen et al., 2016*); the generality and directionality of such relationships in empirical datasets at varying scales of divergence is clearly worthy of further investigation.

The previously cryptic phylogenetic link between cnidarians and placozoans seen in gene sets less influenced by compositional bias will require further testing with other analyses and data modalities, such as rare genomic changes, which should be ever more visible as highly contiguous assemblies continue to be reported from non-bilaterian animals (*Eitel et al., 2018*; *Kamm et al., 2018*;

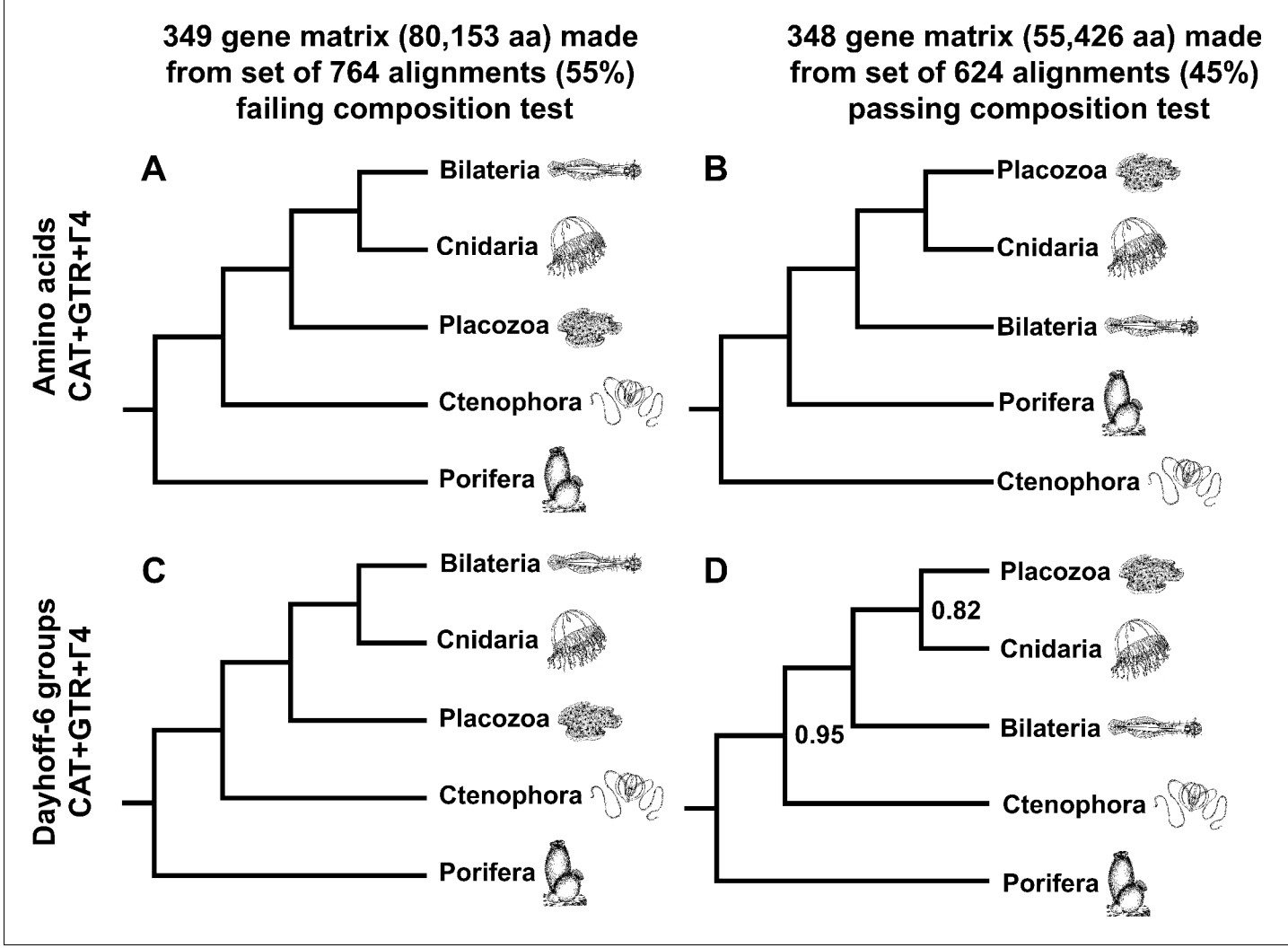

**Figure 4.** Schematic depiction of deep metazoan interrelationships in posterior consensus trees from CAT + GTR + Γ4 mixture model analyses of matrices made from subsets of genes passing or failing a sensitive null-simulation test of compositional heterogeneity. Panels correspond to (A) the amino acid matrix made within the failing set; (B) the amino acid matrix derived from the passing set; (C) the Dayhoff-6 recoded matrix from the failing set; (D) the Dayhoff-6 recoded matrix from the passing set. Only nodes with posterior probability less than 1.00 are annotated numerically.
DOI: https://doi.org/10.7554/eLife.36278.010

The following figure supplements are available for figure 4:

**Figure supplement 1.** Maximum likelihood tree under a profile mixture model inferred from the 349-orthologue matrix composed from the subset of genes binned as failing the null-simulation compositional bias test.
DOI: https://doi.org/10.7554/eLife.36278.011

**Figure supplement 2.** Maximum likelihood tree under a profile mixture model inferred from the 348-orthologue matrix composed from the subset of genes binned as passing the null-simulation compositional bias test.
DOI: https://doi.org/10.7554/eLife.36278.012

*Jiang, 2018*; *Leclère, 2018*). However, if validated, this relationship must continue to raise questions on the homology of certain traits across non-bilaterians. Many workers, citing the incompletely known development (*Eitel et al., 2011*; *Pearse and Voigt, 2007*) and relatively bilaterian-like gene content of placozoans (*Srivastava et al., 2008*; *Eitel, 2017*), presume that these organisms must have a still-unobserved, more typical development and life cycle (*DuBuc et al., 2018*), or else are merely oddities that have experienced wholesale secondary simplification, having scant significance to any evolutionary path outside their own. Indeed, it is tempting to interpret this new phylogenetic position as further bolstering such hypotheses, as much work on cnidarian models in the evo-devo

paradigm is predicated on the notion that cnidarians and bilaterians share, more or less, many homologous morphological features, viz. axial organization (*Genikhovich and Technau, 2017*; *DuBuc et al., 2018*), nervous systems (*Liebeskind et al., 2017*; *Moroz and Kohn, 2016*; *Kelava et al., 2015*; *Kristan, 2016*; *Arendt et al., 2016*), basement-membrane lined epithelia (*Fidler et al., 2017*; *Leys and Riesgo, 2012*), musculature (*Steinmetz et al., 2012*), embryonic germ-layer organisation (*Steinmetz et al., 2017*), and internal digestion (*Presnell et al., 2016*; *Putnam et al., 2007*; *Hejnol and Martindale, 2008*; *Martindale and Hejnol, 2009*). While we do not argue, as some have done (*Schierwater, 2005*; *Syed and Schierwater, 2002*), that placozoans resemble hypothetical metazoan ancestors, we hesitate to dismiss them *a priori* as irrelevant to understanding early bilaterian evolution in particular: although apparently simpler and less diverse, placozoans nonetheless have equal status to cnidarians as an immediate extant outgroup. Rather, we see value in testing assumed hypotheses of homology, character by character, by extending pair-wise comparisons between bilaterians and cnidarians to include placozoans, an agenda which demands reducing the large disparity in embryological, physiological, and molecular genetic knowl-edge between these taxa, towards which recent progress has been made using both established methods such as in situ hybridization (*DuBuc et al., 2018*) and image analysis (*Varoqueaux, 2018*), as well as new technologies such as single-cell RNA-seq (*Sebé-Pedrós, 2018a*; *Sebé-Pedrós et al., 2018b*). Conversely, we emphasize another implication of this phylogeny: characters that can be vali-dated as homologous at any level between Bilateria and Cnidaria must have originated earlier in ani-mal evolution than previously appreciated, and should either cryptically occur in modern placozoans or else have been lost at some point in their ancestry. In this light, paleobiological scenarios of early animal evolution founded on inherently phylogenetically-informed interpretations of Ediacaran fossil forms (*Cavalier-Smith, 2018*; *Cavalier-Smith, 2017*; *Dufour and McIlroy, 2018*; *Sperling and Vinther, 2010*; *Evans et al., 2017*) and molecular clock estimates (*Cunningham et al., 2017*; *dos Reis et al., 2015*; *Dohrmann and Wörheide, 2017*; *Erwin et al., 2011*) may require re-examination.

## Materials and methods

### Sampling, sequencing, and assembling reference genomes from previously unsampled placozoans

Haplotype H4 and H6 placozoans were collected from water tables at the Kewalo Marine Labora-tory, University of Hawaii-Manoa, Honolulu, Hawaii in October 2016. Haplotype H11 placozoans were collected from the Mediterranean '*Anthias*' show tank in the Palma de Mallorca Aquarium, Mal-lorca, Spain in June 2016. All placozoans were sampled by placing glass slides suspended freely or mounted in cut-open plastic slide holders into the tanks for 10 days (*Pearse and Voigt, 2007*). Pla-cozoans were identified under a dissection microscope and single individuals were transferred to 500 µl of RNA*later*, stored as per manufacturer's recommendations.

DNA was extracted from 3 individuals of haplotype H11 and 5 individuals of haplotype H6 using the DNeasy Blood and Tissue Kit (Qiagen, Hilden, Germany). DNA and RNA from three haplotype H4 individuals were extracted using the AllPrep DNA/RNA Micro Kit (Qiagen), with both kits used according to manufacturer's protocols.

Illumina library preparation and sequencing was performed by the Max Planck Genome Centre, Cologne, Germany, as part of an ongoing metagenomics project in marine symbiosis. In brief, DNA/ RNA quality was assessed with the Agilent 2100 Bioanalyzer (Agilent, Santa Clara, USA) and the genomic DNA was fragmented to an average fragment size of 500 bp. For the DNA samples, the concentration was increased (MinElute PCR purification kit; Qiagen, Hilden, Germany) and an Illu-mina-compatible library was prepared using the Ovation Ultralow Library Systems kit (NuGEN, Leek, The Netherlands) according the manufacturer's protocol. For the haplotype H4 RNA samples, the Ovation RNA-seq System V2 (NuGen, 376 San Carlos, CA, USA) was used to synthesize cDNA and sequencing libraries were then generated with the DNA library prep kit for Illumina (BioLABS, Frank-furt am Main, Germany). All libraries were size selected by agarose gel electrophoresis, and the recovered fragments quality assessed and quantified by fluorometry. For each DNA library 14 – 75 million 100 bp or 150 bp paired-end reads were sequenced on Illumina HiSeq 2500 or 4000

machines (Illumina, San Diego, U.S.A); for the haplotype H4 RNA libraries 32 – 37 million single 150 bp reads were obtained.

For assembly, adapters and low-quality reads were removed with bbduk (https://sourceforge.net/projects/bbmap/) with a minimum quality value of two and a minimum length of 36 and single reads were excluded from the analysis. Each library was error corrected using BayesHammer (*Nikolenko et al., 2013*). A combined assembly of all libraries for each haplotype was performed using SPAdes 3.62 (*Bankevich et al., 2012*). Haplotype four and H11 data were assembled from the full read set with standard parameters and kmers 21, 33, 55, 77, 99. The Haplotype H6 data was pre-processed to remove all reads with an average kmer coverage <5 using bbnorm and then assembled with kmers 21, 33, 55 and 77.

Reads from each library were mapped back to the assembled scaffolds using bbmap (https://sourceforge.net/projects/bbmap/) with the option fast = t. Scaffolds were binned based on the mapped read data using MetaBAT (*Kang et al., 2015*) with default settings and the ensemble binning option activated (switch –B 20). The *Trichoplax* host bins were evaluated using metawatt (*Strous et al., 2012*) based on coding density and sequence similarity to the *Trichoplax* H1 reference assembly (NZ_ABGP00000000.1). The bin quality metrics were computed with BUSCO2 (*Simão et al., 2015*) (*Table 1*) and QUAST (*Gurevich et al., 2013*). Both the stringent metagenomics binning procedure (a procedure also expedient in other holobiont organisms (*Celis et al., 2018*)) and the very low proportion of multiple orthologue hits in the BUSCO2 assessment (*Table 1*) attest to the lack of evidence for residual non-placozoan contamination within the scaffolds used for gene prediction.

## Predicting proteomes from transcriptome and genome assemblies

Predicted proteomes from species with published draft genome assemblies were downloaded from the NCBI Genome portal or Ensembl Metazoa in June 2017. For Clade A placozoans, host metagenomic bins were used directly for gene annotation. For the H6 and H11 representatives, annotation was entirely *ab initio*, performed with GeneMark-ES (*Ter-Hovhannisyan et al., 2008*); for the H4 representative, total RNA-seq libraries obtained from three separate isolates (BioProject PRJNA505163) were mapped to genomic contigs with STAR v2.5.3a (*Dobin et al., 2013*) under default settings; merged bam files were then used to annotate genomic contigs and derive predicted peptides with BRAKER v1.9 (*Hoff et al., 2016*) under default settings. Choanoflagellate proteome predictions (*Simion et al., 2017*) were provided as unpublished data from Dan Richter. Peptides from a *Calvadosia* (previously *Leucosolenia*) *complicata* transcriptome assembly were downloaded from compagen.org. Peptide predictions from *Nemertoderma westbladi* and *Xenoturbella bocki* as used in Cannon et al 2016 (*Cannon et al., 2016*) were provided directly by the authors. The transcriptome assembly (raw reads unpublished) from *Euplectella aspergillum* was provided by the Satoh group, downloaded from (http://marinegenomics.oist.jp/kairou/viewer/info?project_id=62). Predicted peptides were derived from Trinity RNA-seq assemblies (multiple versions released 2012–2016) as described by Laumer et al (*Laumer et al., 2015*). for the following sources/SRA accessions:: Porifera: *Petrosia ficiformis*: SRR504688, *Cliona varians*: SRR1391011, *Crella elegans*: SRR648558, *Corticium candelabrum*: SRR504694-SRR499820-SRR499817, *Spongilla lacustris*: SRR1168575, *Clathrina coriacea*: SRR3417192, *Sycon coactum*: SRR504689-SRR504690, *Sycon ciliatum*: ERR466762, *Ircinia fasciculata*, *Chondrilla caribensis* (originally misidentified as *Chondrilla nucula*) and *Pseudospongosorites suberitoides* from (https://dataverse.harvard.edu/dataverse/spotranscriptomes); Cnidaria: *Abylopsis tetragona*: SRR871525, *Stomolophus meleagris*: SRR1168418, *Craspedacusta sowerbyi*: SRR923472, *Gorgonia ventalina*: SRR935083; Ctenophora: *Vallicula multiformis*: SRR786489, *Pleurobrachia bachei*: SRR777663, *Beroe abyssicola*: SRR777787; Bilateria: *Limnognathia maerski*: SRR2131287. All other peptide predictions were derived through transcriptome assembly as paired-end, unstranded libraries with Trinity v2.4.0 (*Haas et al., 2013*), running with the –trimmomatic flag enabled (and all other parameters as default), with peptide extraction from assembled transcripts using TransDecoder v4.0.1 with default settings. For these species, no ad hoc isoform selection was performed: any redundant isoforms were removed during tree pruning in the orthologue determination pipeline (see below).

## Orthologue identification and alignment

Predicted proteomes were grouped into top-level orthogroups with OrthoFinder v1.0.6 (*Emms and Kelly, 2015*), run as a 200-threaded job, directed to stop after orthogroup assignment, and print grouped, unaligned sequences as FASTA files with the '-os' flag. A custom python script ('renamer. py') was used to rename all headers in each orthogroup FASTA file in the convention [taxon abbreviation] + '@' + [sequence number as assigned by OrthoFinder SequenceIDs.txt file], and to select only those orthogroups with membership comprising at least one of all five major metazoan clades plus outgroups, of which exactly 4300 of an initial 46,895 were retained. Scripts in the Phylogenomic Dataset Construction pipeline (*Yang and Smith, 2014*) were used for successive data grooming stages as follows: Gene trees for top-level orthogroups were derived by calling the fasta_to_tree.py script as a job array, without bootstrap replicates; six very large orthogroups did not finish this process. In the same directory, the trim_tips.py, mask_tips_by_taxonID_transcripts.py, and cut_long_internal_branches.py scripts were called in succession, with './. tre 10 10', './.y', and './. mm 1 20. /' passed as arguments, respectively. The 4267 subtrees generated through this process were concatenated into a single Newick file and 1419 orthologues were extracted with UPhO (*Ballesteros and Hormiga, 2016*). Orthologue alignment was performed using the MAFFT v7.271 'E-INS-i' algorithm, and probabilistic masking scores were assigned with ZORRO (*Wu et al., 2012*), removing all sites in each alignment with scores below five as described previously (*Laumer et al., 2015*). 31 orthologues with retained lengths less than 50 amino acids were discarded, leaving 1388 well-aligned orthologues.

## Matrix assembly

A full concatenation of all retained 1388 orthogroups was performed with the 'geneStitcher.py' script distributed with UPhO available at https://github.com/ballesterus/PhyloUtensils. However, such a matrix would be too large for tractably inferring a phylogeny under well-fitting mixture models such as CAT + GTR; therefore we used MARE v0.1.2 (*Misof et al., 2013*) to extract an informative subset of genes using tree-likeness scores, running with '-t 100' to retain all taxa and using '-d 1' as a tuning parameter on alignment length. This yielded our 430-orthologue, 73,547 site matrix, with a mean partition length of 202.24 (s.d. 116.96) residues.

As a check on the above procedure, which is agnostic to the identity of the genes assigned into orthologue groups, we also sought to construct a matrix using complete, single-copy sequences identified by the BUSCO v3.0.1 algorithm (*Simão et al., 2015*), using the 303-gene eukaryote_odb9 orthologue set. BUSCO was run independently on each peptide FASTA file used as input to OrthoFinder, and a custom python script ('extract.py') was used to parse the full output table from each species, selecting only those entries identified as complete-length, single-copy representatives of each BUSCO orthologue, and grouping these into unix directories, facilitating downstream alignment, probabilistic masking, and concatenation, as described for the OrthoFinder matrix. This 303-gene BUSCO matrix had a total length of 94,444 amino acids, with 39.6% of sites representing gaps or missing data, with mean partition length 311.70 (standard deviation 202.78).

Within the gene bins nominated by the test of compositional heterogeneity (see below), matrices were constructed again by concatenating and reducing matrices with MARE, using '-t 100' to retain all taxa and setting '-d 0.5' to yield a matrix of an optimal size for inferring a phylogeny under the CAT + GTR model. This procedure gave a 349-gene matrix of 80,153 amino acids (mean partition lengths 228.67 ± s.d. 136.19, 41.64% gaps) within the test-failing gene set, and a 348-gene matrix of 55,426 amino acids (mean partition lengths 158.27 ± s.d. 79.06, 38.92% gaps), within the test-passing set (*Figure 4*).

## Phylogenetic inference

Individual ML gene trees were constructed on all 1388 orthologues in IQ-tree v1.6beta, with '-m MFP -b 100' passed as parameters to perform automatic model selection and 100 standard nonparametric bootstraps on each gene tree.

For inference on the initial 430-gene matrix, we proceeded as follows: ML inference on the concatenated matrix (*Figure 1—figure supplement 1*) was performed with IQ-tree v1.6beta, passing '-m C60 +LG + FO + R4 bb 1000' as parameters to specify a profile mixture model and retain 1000 trees for ultrafast bootstrapping; the '-bnni' flag was used to incorporate NNI correction during

UF bootstrapping, an approach shown to control misleading inflated support arising from model misspecification (*Hoang et al., 2018*). ML inference using only the H1 haplotype as a representative of Placozoa (*Figure 1—figure supplement 2*) was undertaken similarly, albeit using a marginally less complex profile mixture model (C20 +LG + FO + R4). Bayesian inference under the CAT + GTR + Γ4 model was performed in PhyloBayes MPI v1.6j (*Lartillot et al., 2013*) with 20 cores each dedicated to four separate chains, run for 2885–3222 generations with the '-dc' flag applied to remove constant sites from the analysis, and using a starting tree derived from the FastTree2 program (*Price et al., 2010*). The two chains used to generate the posterior consensus tree summarized in *Figure 1* converged on exactly the same tree in all MCMC samples after removing the first 2000 generations as burn-in. Analysis of Dayhoff-6-state recoded matrices in CAT + GTR + Γ4 was performed with the serial PhyloBayes program v4.1c, with '-dc -recode dayhoff6' passed as flags. Six chains on the 430- gene matrix were run from 1441 to 1995 generations; two chains showed a maximum bipartition discrepancy (maxdiff) of 0.042 after removing the first 1000 generations as burn-in (*Figure 2*). QuartetScores (*Zhou, 2017*) was used to measure internode certainty metrics including the reported EQP-IC, using the 430 gene trees from those orthologues used to derive the matrix as evaluation trees, and using the amino-acid CAT + GTR + Γ4 tree as the reference to be annotated (*Figure 1*).

For inference on the BUSCO 303 gene set, we ran 4 chains of the CAT + GTR + Γ4 mixture model with PhyloBayes MPI v1.7a, applying the -dc flag again to remove constant sites, but here not specifying a starting tree; chains were run from 1873 to 2361 generations. Unfortunately, no pair of chains reached strict convergence on the amino-acid version of this matrix (with all pairs showing a maxdiff = 1 at every burn-in proportion examined), perhaps indicating problems mixing among the four chains we ran. However, all chains showed full posterior support for identical relationships among the five major animal groups, with differences among chains assignable to minor differences in the internal relationships within Choanoflagellata and Bilateria. Accordingly, the posterior consensus tree in *Figure 3A* is summarized from all four chains, with a burn-in of 1000 generations, sampling every 10 generations. For the Dayhoff-recoded version of this matrix, we ran six separate chains again with CAT + GTR + Γ4 with the -dc flag, for 5433 – 6010 generations; two chains were judged to have converged, giving a maxdiff of 0.141157 during posterior consensus summary with a burn-in of 2500, sampling every 10 generations (*Figure 3B*).

For inference on the 348 and 349 gene matrices produced within gene bins defined by the null-simulation test of compositional bias (see below), we ran six chains each for the amino acid and recoded versions of each matrix, under CAT + GTR + Γ4 with constant sites removed. In the amino-acid matrix, chains ran from 2709 to 3457 and 1423 – 1475 generations for the test-failing and test-passing matrices, respectively. In the recoded matrix, chains ran from 3893 to 4480 and 4350 – 4812 generations for the test-failing and test-passing matrices, respectively. In selecting chains to input for posterior consensus summary tree presentation (*Figure 4A–D*), we chose pairs of chains and burn-ins that yielded the lowest possible maxdiff values (all <0.1 with the first 500 generations discarded as burn-in, except for the amino-acid coded test-failing matrix, whose most similar pair of chains gave a maxdiff of 0.202 with 1000 generations discarded as burn-in). We emphasize that the topologies and supports displayed in *Figure 4A–D* are similar when all chains (and conservative burn-in values) are used to generate consensus trees. For ML trees using profile mixture models for the test-failing (*Figure 4—figure supplement 1*) and test-passing (*Figure 4—figure supplement 2*) gene matrices, we used IQ-tree 1.6rc, calling in the same manner (with C60 +LG + FO + R4) as used on our 430-gene matrix (see above).

## Tests of compositional heterogeneity

For posterior predictive tests of compositional heterogeneity and residue diversity using MCMC samples under CAT + GTR (*Table 2*), we used PhyloBayes MPI v1.8 to test two chains from the initial 430-gene matrix, three chains from the 303-gene BUSCO matrix, and six chains each from the 348 (test-failing) and 349 (test-passing) gene matrices, removing 2000 generations from the first matrix and 1000 from the others as burn-in. Results from tests on representative chains were selected for plotting in *Figure 3C* and summary in *Table 2*; however, results from all chains tested are deposited in the Data Dryad accession.

For the per-gene null simulation tests of compositional bias (*Foster, 2004*), we used the p4 package (https://github.com/pgfoster/p4-phylogenetics), inputting the ML trees inferred by IQ-tree for

each of the 1388 alignments, and assuming an LG+$\Gamma$4 substitution model with a single empirical frequency vector for each gene; this test was implemented with a simple wrapper script ('p4_compo_test_multiproc.py') leveraging the python multiprocessing module. We opted not to model-test each gene individually in p4, both because the range of models implemented in p4 are more limited than those tested for in IQ-tree, and because, as a practical matter, LG (usually with variant of the FreeRates model of rate heterogeneity) was chosen as the best-fitting model in the IQ-tree model tests for a large majority of genes, suggesting that LG+$\Gamma$4 would be a reasonable approximation for the purposes of this test. We selected an $\alpha$-threshold of 0.10 for dividing genes into test-passing and -failing bins as a conservative measure; however, we emphasize that even at a less conservative $\alpha$ = 0.05, 47% of genes would still be detected as falling outside the null expectation.

## Source data availability

SRA accession codes, where used, and all alternative sources for sequence data (e.g. individually hosted websites, personal communications), are listed above in the Materials and methods section. A DataDryad accession is available at https://doi.org/10.5061/dryad.6cm1166, which makes available all helper scripts, orthogroups, multiple sequence alignments, phylogenetic program output, and raw host proteomes inputted to OrthoFinder. Metagenomic bins containing placozoan host contigs and gene annotations from H4, H6 and H11 isolates are also provided in this accession. PhyloBayes. chain files, due to their large size, are separately accessioned at in Zenodo at https://doi.org/10.5281/zenodo.1197272.

## Acknowledgements

Nicole Dubilier (Max Planck Institute for Marine Microbiology) contributed resources that permitted the collection and assembly of draft *Trichoplax* genomes, which were amplified and sequenced at the Max Planck-Genome-Centre Cologne. Dan Richter (King lab) and Kanako Hisata (Satoh lab) provided access to unpublished transcriptomes and peptide predictions. The EMBL-EBI Systems Infrastructure team provided essential support on the EBI compute cluster. Allen Collins, Scott Nichols, and particularly Andreas Hejnol provided useful comments on an earlier version of this manuscript.

## Additional information

### Funding

| Funder | Grant reference number | Author |
|---|---|---|
| Max-Planck-Institut fuer Marine Microbiologie | | Harald Gruber-Vodicka |
| European Bioinformatics Institute | | John C Marioni |
| Harvard University | Faculty of Arts and Sciences | Gonzalo Giribet |

The funders had no role in study design, data collection and interpretation, or the decision to submit the work for publication.

### Author contributions

Christopher E Laumer, Conceptualization, Resources, Data curation, Software, Formal analysis, Validation, Investigation, Visualization, Methodology, Writing—original draft, Project administration, Writing—review and editing; Harald Gruber-Vodicka, Resources, Formal analysis, Supervision, Investigation, Writing—review and editing; Michael G Hadfield, Resources, Writing—review and editing; Vicki B Pearse, Conceptualization, Resources, Writing—review and editing; Ana Riesgo, Resources, Data curation, Writing—review and editing; John C Marioni, Resources, Supervision, Writing—review and editing; Gonzalo Giribet, Conceptualization, Resources, Funding acquisition, Writing—review and editing

## Author ORCIDs

Christopher E Laumer  http://orcid.org/0000-0001-8097-8516
John C Marioni  http://orcid.org/0000-0001-9092-0852

## Decision letter and Author response

Decision letter https://doi.org/10.7554/eLife.36278.021
Author response https://doi.org/10.7554/eLife.36278.022

## Additional files

### Supplementary files

• Transparent reporting form
DOI: https://doi.org/10.7554/eLife.36278.013

### Data availability

SRA accession codes, where used, and all alternative sources for sequence data (e.g. individually hosted websites, personal communications), are listed above in the Materials and Methods section. A DataDryad accession is available at https://doi.org/10.5061/dryad.6cm1166, which makes available all helper scripts, orthogroups, multiple sequence alignments, phylogenetic program output, and raw host proteomes inputted to OrthoFinder. Metagenomic bins containing placozoan host contigs and raw RNA reads used to derive gene annotations from H4, H6 and H11 isolates are also provided in this accession. PhyloBayes. chain files, due to their large size, are separately accessioned at in Zenodo at https://doi.org/10.5281/zenodo.1197272. Raw RNA-seq data from three separate isolates of Placozoan haplotype H4 have been deposited in NCBI BioProject under accession code PRJNA505163.

The following datasets were generated:

| Author(s) | Year | Dataset title | Dataset URL | Database and Identifier |
|---|---|---|---|---|
| Laumer C, Gruber-Vodicka H, Hadfield MG, Pearse VB, Riesgo A, Marioni JC, Giribet G | 2018 | Data from: Placozoa and Cnidaria are sister taxa | https://doi.org/10.5061/dryad.6cm1166 | Dryad Digital Repository, 10.5061/dryad.6cm1166 |
| Laumer C, Gruber-Vodicka H, Hadfield MG, Pearse VB, Riesgo A, Marioni JC, Giribet G | 2018 | PhyloBayes chain data from: Placozoa and Cnidaria are sister taxa | https://doi.org/10.5281/zenodo.1197272 | Zenodo, 10.5281/zenodo.1197272 |
| Christopher E Laumer, Harald Gruber-Vodicka, Michael G Hadfield, Vicki B Pearse, Ana Riesgo, John C Marioni, Gonzalo Giribet | 2018 | Amplified total RNA from Placozoan H4 isolates | https://www.ncbi.nlm.nih.gov/bioproject/PRJNA505163 | NCBI BioProject, PRJNA505163 |

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
