## [Decision Letter]

Thank you for submitting your article "Placozoa and Cnidaria are sister taxa" for consideration by *eLife*. Your article has been reviewed by Diethard Tautz as the Senior Editor, a Reviewing Editor and three reviewers. The following individuals involved in review of your submission have agreed to reveal their identity: Davide Pisani (Reviewer #2); David C Plachetzki (Reviewer #3).

The reviewers have discussed the reviews with one another and the Reviewing Editor has drafted this decision to help you prepare a revised submission

Summary:

The manuscript by Laumer and colleagues is the latest chapter in the quest to understand (and question) relationships at the base of the animal tree. For quite some time, the controversy has focused on whether poriferans or ctenophores are the sister lineage to the rest of the metazoan phyla, whereas the rest of the relationships appear to have been somewhat stable toward a grouping of bilaterians + cnidarians, with placozoans as the sister to that lineage. This study questions the validity of this later set of accepted relationships as well as sheds additional light on the controversy about sponges-sister vs. ctenophores-sister. The central claim made by Laumer et al., is that among lineage compositional heterogeneity has plagued most previous analyses of similar phylogenetic depth and that the finding of bilateria + cnidaria to the exclusion of placozoa is the result of such error. Such a conclusion would force a reevaluation of the significance of placozoans in our understanding of early metazoan evolution and would additionally place our understanding of placozoan biology in a new context.

Essential revisions:

1) New placozoan genomes were sequenced and assembled in this paper, but there is very little description of the properties of these new assemblies. Several questions related to the contiguity of the assemblies produced, how the new assemblies compare with the existing assembly and gene models for *Trichoplax adhaerens*, the degree of overlap among the new genome assemblies reported here, BUSCO scores, etc., of the protein models produced from these analyses, are not addressed. Similarly, the phylogenomic data matrix lacks descriptions of key attributes such as global taxon occupancy and a distribution of partition lengths. Finally, one challenge when sequencing and assembling genomes and, to a lesser extent transcriptomes, from marine meiofauna is contamination. It was not clear how contamination was dealt with in the preparation of genome assemblies, or in the phylogenomic dataset construction procedure. The central claim of the paper is that compositional heterogeneity drives a certain analytical outcome in this and in previous analyses, but given the limited description of the data that underlie this result, one could imagine other artifacts, unrelated to compositional heterogeneity, that could also be at play.

2) The paper leverages new genome scale datasets for placozoa, but it is not clear how the existing genome for *Trichoplax adhaerens* was utilized. Our reading is that a sample of Haplotype H1, the same haplotype that has been previously sequenced, was assembled here and data from this new H1 assembly was utilized, together with separate assemblies of the other haplotypes. If this is correct, it is not clear why the existing high-quality protein models for *Trichoplax* were not utilized in the production of the phylogentic matrix. This simple inclusion could help allay concerns over the nature of the new data.

3) Laumer et al., propose that the findings of the majority of previous analyses were influenced by compositional heterogeneity, as uncovered in their analyses. We note that the effects of compositional heterogeneity in the context of deep metazoan phylogenomics were explored previously at least once previously (in Borowiec et al., 2015) using dahoff 6 recoding; however, the Borowiec et al., study did not find the result reported here using the existing *Trichoplax* data. Moreover, the effect of compositional heterogeneity in previous studies were not directly tested by Laumer et al. Taxon sampling and extensive differences in the datasets utilized used in previous analyses are additional variables that need to be accounted for. At least a subset of previously published phylogenomic datasets that bear on the position of placozoa should be reanalyzed under the procedure used here, and cases where this has already been done should be addressed.

4) Results and Discussion section. The reported PPA scores are very bad, indicating that both datasets are extremely heterogeneous and that at the AA level both datasets are unreliable. The PPA do suggest BUSCO is worst, but that does not make your new AA dataset more reliable as Z-scores greater than one hundred indicate an utter failure of the model to describe the data (see Feuda et al., 2017 for a discussion of interpretation of Z-scores). This should be pointed out in your discussion. Note however, that the fact that a dataset is highly heterogeneous does not mean that the tree it supports is incorrect; that depends on the specific of the dataset (taxon specific distribution of heterogeneity, presence of other forms of noise etc.). High heterogeneity means that we should be cautious when interpreting the results as the tree might include tree-reconstruction artefacts. So, our reading of these two analyses is that both datasets are heterogeneous and that tests (like analyses under Dayhoff-6) need to be carried out to validate the clades in these trees. You did these tests and they clearly indicate that Placo-Cni and Pori-sister are unlikely to be compositional artefact while Cneto-sister is likely a compositional artefact.

5) Results and Discussion section: You need to provide more information about heterogeneity. Are you talking of site or lineage specific heterogeneity? These are two different components of a dataset’s heterogeneity that are differently modeled and both need to be considered. Site-specific heterogeneity is generally modelled relatively efficiently using CAT-GTR while lineage specific heterogeneity cannot be modeled using the standard models you used in this paper, as you would need BreakpointCAT, NDCH or similar to model it. D6 Recoding reduces both forms of heterogeneity and, in combination with CAT-GTR generally, allows for an adequate modeling of site-specific heterogeneity and an improvement in the modeling of lineage specific heterogeneity (the latter not necessarily reaching adequacy – see Feuda et al., 2017). Hence it is key that you report statistics for all your datasets (Z-scores for both site and lineage-specific heterogeneity for all you're a and Dayhoff datasets). A table with the Diversity and Comp PPA (these can be derived using Phylobayes) will do and should be easily done as you should already have all the chains (as you already run other PPA). A table presenting all z cores for all your dataset would be key to interpret your results (Figure 1 to Figure 4D included) and decide which tree is less likely to be incorrect/more likely to be accurate.

6) Results and Discussion section: This section reads like an ad hoc attempt to justify Ctenophora-sister, a result that is tangential to your paper (which is about Placozoa) and that is not supported by your analyses. Specifically:

6A) Figure 3 presents the BUSCO trees under CAT-GTR+G (3A) is AA and (3B) is Dayhoff. Figure 3A (your worst overall dataset in terms of heterogeneity) clearly supports Ctenophora-Sister, while Figure 3B shows an unorthodox PORIFERA+CTENOPHORA (but with relatively low support ~ 75%). Neither of these two figures supports Porifera-sister. According to your arguments BUSCO is the worst dataset. When considered as an AA alignment, the BUSCO dataset maximizes its compositional heterogeneity and supports Cteno-sister. In the text you state that Figure 4A (the dataset of the heterogeneous genes in your dataset – that we will discuss further below) provides evidence that Pori-sister might be linked to heterogeneity. However, even if that was actually the case, then Figure 3A must be evidence that Cteno-sister is a compositional artefact (equally linked to heterogeneity). Yet this important fact is not even mentioned in the text. This is surprising and unfair given that you use the heterogeneity of the dataset used to derive Figure 4A to make an argument against Porifera-sister, the heterogeneity of Figure 3A to suggest Placozoa + Bilateria is likely to be an artefact, but you do not discuss the fact that also Cteno-sister emerges from the "bad" BUSCO dataset and should thus be considered to be dubious. The text needs to be modified to reflect this.

6B) The experiment in Figure 4 is nicely designed, but some aspects of its implementation and the interpretation of its results are problematic. Specifically, we think that the analyses reported in Figure 4A and 4C are misleading for two reasons. First, the data matrix cannot be proved to be composed only by heterogeneous genes. This is because 111 of the 764 genes in the dataset used to derive this figure (~ 14% of the superalignment) are composed of genes that are not, strictly speaking (i.e., at the 0.05 cutoff level), heterogeneous. Second, even if this dataset was composed purely of heterogeneous genes it could not be used to claim, for example, that Porifera-sister is linked to heterogeneity. This is because on this dataset Dayhoff recoding did not change the position of the sponges. This indicates that the signal for this node in this dataset (irrespective of its heterogeneity) is not driven by the within-D6classes changes that are known to be associated with compositional heterogeneity. Irrespective of the heterogeneity of the dataset, the fact that Porifera sister is unchanged and does not loose support upon recoding indicates that it is supported by between D6-classes changes, which are not silenced by recoding, and are more likely to be associated phylogenetic signal. To conclude, you do not have evidence in Figure 4A and 4C to claim that the assertion of Feuda (that Cteno-sister is likely to be associated with compositional heterogeneity while Porifera-sister is not likely to be a compositional artifact) might be incorrect. The Dayhoff test rejects your conclusion. So, what to do with the analysis of Figure 4A and 4C? Our thought is that this dataset is confusing your experiments because it is composed of a mixture of more and less heterogeneous genes and it is best excluded. We suggest that you just report the results in Figure 4B and 4D which are actually very clear and say all that there is to say on Placo-Cni and Pori-sister vs. Cteno-sister.

6C) The fact that the genes in your "homogeneous dataset" are individually homogeneous does not mean that your homogenous superalignment is in itself homogeneous. This needs to be tested (using PPA under CAT-GTR for both Dayhoff-6 and AA datasets). This is because heterogeneity in a multigene dataset adds up and it is customary that superalignments composed of individually homogeneous genes fail heterogeneity tests. My expectation is that the dataset in Figure 4B will be shown to be still heterogeneous and that of Figure 4D will be shown to be much more homogeneous (as it is always the case with D6-recoded datasets). Accordingly, the analyses of Figure 4BD indicate that there is evidence for Cteno-sister to be a compositional artefact (driven by within D6-classes changes only present in Figure 4B) while there is no evidence for Cnidaria+Placoza and Pori-sister to be compositional artifacts (as they are both present in Figure 4D where only between classes changes are evident). In addition, this analysis confirms the existence of two signals in the data with reference to Ctenophores (one linked to unreliable within D6 classes changes) and one linked to the more reliable between classes changes, while Placo-Cnidaria seem to be the only signal in the dataset and it is more strongly represented in more reliable between-classes changes.

7) Results and Discussion section: Here you seem to assume that Dayhoff recoding always have to change the topology and that if the topology does not change the recoding, in some way, it failed. The application of D6 was "not significant" (in your words) for the dataset of Figure 4AC. The problem with your statement is that a tree does not need to be incorrect simply because the data are heterogeneous, hence D6 does not need to invariably cause topological changes. So, its application was not "not significant", but it did not drive to a topological change. As said above as well, D6 reduces the influence of compositional heterogeneity by silencing within category changes. The analysis in Figure 4AC indicates that Porifera-sister is driven by more reliable among category changes which are generally not compositionally driven. Otherwise it would have disappeared in the Dayhoff analysis. Note that we are not saying that Pori-sister is correct; rather, we are saying that Figure 4A C does not provide evidence that this clade is compositionally driven, hence you cannot conclude that Porifera-sister might be associated with compositional attractions. Note that, with reference to the Ctenophora debate, the application of recodings has invariably produced strongly directional changes Ctenophora->Porifera. This is a highly non-random directional change and at this time there is no known Dataset that while supporting Porifera-sister at the AA level (under CAT-GTR) was found to support Cteno-sister after recoding. This is true of all the datasets of Whelan when Choanoflagellata are the only outgroups used (pori-sister as AA and as Dayhoff), and for your dataset in Figure 4AC – for example. This can only be interpreted as suggesting that in all available datasets Porifera-sister is never driven by within category changes, accordingly, when it emerges at the root, is cannot be because of a compositional attraction – irrespective of the heterogeneity of the dataset itself. Differently, D6 experiments invariably indicate that when ctenophores emerges at the base, compositional heterogeneity can never (all datasets tested to date behave exactly in this same way -including yours) be ruled out – strongly suggesting Cteno-sister to be a compositional artifact.

8) Title: Given how controversial the relationships at the base of the animal tree have been, perhaps a more moderate title would be more fitting (e.g., Phylogenomic support for a sister relationship of Placozoa and Cnidaria).

9) The scripts used for data processing in genome assembly, phylogenomics matrix construction and analysis are not linked in the paper. Sharing these scripts on a publicly available repository like Github would enhance the reproducibility of the present study.

[Editors' note: further revisions were requested prior to acceptance, as described below.]

Thank you for submitting your article "Support for a clade of Placozoa and Cnidaria in genes with minimal compositional bias" for consideration by *eLife* and for your patience during the second round of reviewers. Your article has been re-reviewed and the evaluation has been overseen by a Reviewing Editor and Diethard Tautz as the Senior Editor. The following individual involved in review of your submission has agreed to reveal his identity: David C Plachetzki (Reviewer #3).

The reviewers have discussed the reviews with one another and the Reviewing Editor has drafted this decision to help you prepare a revised submission.

Summary:

The manuscript was markedly improved and the authors' efforts are greatly appreciated. However, the concern remains that something more than compositional bias lies behind Cnidaria+Placozoa and that this finding is unstable at best. Three critical pieces of information emerge from the revision and the authors’ response: (1) Compositionally unbiased partitions that support Cnidaria+Placozoa are systematically shorter than partitions that do not support this clade. (2) A reanalysis of the Simeon et al., dataset, which is much larger, did not support Cnidaria+Placozoa following a similar phylogenetic approach used by Laumer et al. (3) In unpublished data, Cnidaria+Placozoa is only recovered in analyses that lack specific outgroups. Together, these findings suggest that something other than the removal of compositional bias, particular to the Laumer et al., dataset, is driving Cnidaria+Placozoa.

Essential revisions:

1) The authors allay some concerns over the description of the genome data and the point that the new genome assemblies produce BUSCO orthologue occupancies that are similar to other transcriptome assemblies is appreciated. One does wonder why short read genome assemblies, which produced highly fragmented assemblies, were utilized in the first place, instead of transcriptomes. If the purpose of doing so was to enhance placozoan genome resources, these resources are not described here in any detail because they are too fragmented. "we opted not [sic] to omit a detailed characterisation of these genomes, partially out of spatial constraints (this article was submitted as a Short Report format), and partially since the genomes themselves are Illumina short-insert-only assemblies and therefore relatively fragmented". Please provide this information.

2) The manuscript now includes information of the percentage, means and standard deviations for partition lengths. This information is important for evaluating the possible causes of the phylogenetic discrepancies (e.g. Cnidaria+Placozoa vs. Cnidaria+Bilateria) observed in different datasets. We now see that dataset that supports Cnidaria+Placozoa are systematically composed of shorter partitions. Thus, compositional bias is not the only difference between the two datasets. No further explanation is given. Please report these data and explicitly discuss the finding that ompositional bias is not the only difference between the two datasets.

3) The paragraph describing metaBAT and metawatt does not explicitly mention contamination, which would help the reader interpret the purpose of these steps. Please rectify this.

4) We agree that Dayhoff-6 recoding is not a silver bullet for compositional bias and that previous analyses that have not filtered data for compositionally unbiased partitions are not direct comparisons. However, the re-analysis of the Simeon dataset that the authors conducted, where unbiased partitions were selected and analyzed in both amino acid and Dayhoff-6 recoded space, does represent a more or less direct comparison and these results strongly contradict the topology favored by Laumer et al. While there are expectedly differences in taxon sampling and data handling between the Simeon re-analysis and that presented by Laumer et al., but this is true of the vast majority of previous phylogenomic analyses at this scale which confidently resolve other nodes in the animal tree of life. The explanation for this in the author's response, "this published dataset does not contain any signal for a Cnidaria-Placozoa relationship" seems quite correct indeed, but phylogenomic datasets as large as Simeon (which is much larger than that presented by Laumer et al.) should possess such phylogenetic signal if it were robust. Therefore, this re-analysis of the Simeon dataset, which is not included in the revision, draws the central phylogenetic finding of Laumer et al., into question and strongly suggests that some other feature of the Laumer et al., dataset is contributing to Cnidaria+Placozoa. As stated above, the unbiased partitions are significantly shorter than the biased partitions. It is possible that this finding is somehow, perhaps indirectly, related to the central finding?

5) Again, while Ctenophora-vs.-Porifera is tangential to the central claims of the paper, I did find it relevant that in the authors’ response (unpublished data), Cnidaria+Placozoa was recovered in "one" analysis, but only when certain outgroups were removed, and the data were D6 recoded. This finding, like the Simeon reanalysis, speaks strongly to the lability of Cnidaria+Placozoa. Please comment on this issue.

---

## [Author Response]

Essential revisions:1) New placozoan genomes were sequenced and assembled in this paper, but there is very little description of the properties of these new assemblies. Several questions related to the contiguity of the assemblies produced, how the new assemblies compare with the existing assembly and gene models for Trichoplax adhaerens, the degree of overlap among the new genome assemblies reported here, BUSCO scores, etc., of the protein models produced from these analyses, are not addressed. Similarly, the phylogenomic data matrix lacks descriptions of key attributes such as global taxon occupancy and a distribution of partition lengths. Finally, one challenge when sequencing and assembling genomes and, to a lesser extent transcriptomes, from marine meiofauna is contamination. It was not clear how contamination was dealt with in the preparation of genome assemblies, or in the phylogenomic dataset construction procedure. The central claim of the paper is that compositional heterogeneity drives a certain analytical outcome in this and in previous analyses, but given the limited description of the data that underlie this result, one could imagine other artifacts, unrelated to compositional heterogeneity, that could also be at play.

Although this article does indeed introduce new draft genomic assemblies from a hitherto-unsequenced lineage of Placozoa, we opted not to omit a detailed characterisation of these genomes, partially out of spatial constraints (this article was submitted as a Short Report format), and partially since the genomes themselves are Illumina short-insert-only assemblies and therefore relatively fragmented. However, because of the generally small apparent intron size of Placozoa, even such fragmentary assemblies show an acceptable degree of completeness (in terms of gene content) similar or superior to many contemporary transcriptome assemblies, judging by BUSCO orthologue occupancy. The reviewers make a good point that some basic summary statistics on these new draft assemblies would improve the paper; we therefore now include Table 1 to this end.

We somewhat contest the comment that the matrix contents are poorly described, see e.g. the first paragraph of the Results and Discussion section (and the bar plot in Figure 1 depicting matrix occupancy for each species). Nonetheless, following this critique we now include in the Materials and Methods a few more general summary statistics for each matrix described in the paper (global gap% age, means and standard deviations for partition lengths). One curious trend, also apparent from total supermatrix length, is that the supermatrices in which we have found support for Cnidaria+Placozoa seem to be systematically composed of shorter partitions, suggesting possibly that longer orthologues are more susceptible to compositional variation over large evolutionary timescales.

We were surprised by the comment that “It was not clear how contamination was dealt with in the preparation of genome assemblies”, and the apparent insinuation that our results may be explicable as the result of contamination. In the last paragraph of the first Materials and Methods section on sampling, we clearly describe a metagenome binning procedure used to extract putative host bins (metaBAT) apart from contaminating symbiotic bacterial genomes. The low BUSCO duplication rate in each assembly also attests to the lack of evidence for contaminating eukaryotic contigs being present in our draft metagenome assembly bins. To the extent that contamination may have been present in other libraries used in our orthology analysis, we cannot comment specifically, although we do observe that if contamination was indeed the explanation for our major result, one would in general expect to see Placozoa forming the sister group to one specific contaminating cnidarian lineage, not the mutual monophyly we have recovered.

2) The paper leverages new genome scale datasets for placozoa, but it is not clear how the existing genome for Trichoplax adhaerens was utilized. Our reading is that a sample of Haplotype H1, the same haplotype that has been previously sequenced, was assembled here and data from this new H1 assembly was utilized, together with separate assemblies of the other haplotypes. If this is correct, it is not clear why the existing high-quality protein models for Trichoplax were not utilized in the production of the phylogentic matrix. This simple inclusion could help allay concerns over the nature of the new data.

As explained in the caption for Figure 1 in the reviewed submission, the terminal taxon we label haplotype H1 is the reference assembly from Srivastava et al., 2008.

3) Laumer et al., propose that the findings of the majority of previous analyses were influenced by compositional heterogeneity, as uncovered in their analyses. We note that the effects of compositional heterogeneity in the context of deep metazoan phylogenomics were explored previously at least once previously (in Borowiec et al., 2015) using dahoff 6 recoding; however, the Borowiec et al., study did not find the result reported here using the existing Trichoplax data. Moreover, the effect of compositional heterogeneity in previous studies were not directly tested by Laumer et al. Taxon sampling and extensive differences in the datasets utilized used in previous analyses are additional variables that need to be accounted for. At least a subset of previously published phylogenomic datasets that bear on the position of placozoa should be reanalyzed under the procedure used here, and cases where this has already been done should be addressed.

Our Figure 3 and Figure 4A/4C show that Dayhoff-6 recoding is not sufficient to rescue Placozoa+Cnidaria in at least some compositionally biased datasets. Dayhoff-6 recoding is able to mask only a subset of potential compositionally driven changes; for a more detailed discussion of this, see the recent commentary paper by Laumer in Integrative & Comparative Biology. Therefore, if the matrix constructed for the Borowiec et al., 2015 study was compromised by compositional bias in a similar manner as the matrices we analysed in Figure 3 and Figure 4A/4C, it's not surprising that the authors of this paper might not have recovered Cnidaria+Placozoa either, even with Dayhoff-6 recoding.

It is not clear to us what is meant by the comment “Moreover, the effects of compositional heterogeneity in previous studies were not directly tested by Laumer et al.,” – a more specific rephrasing of this point would be required for us to respond meaningfully.

The reviewer’s request to reanalyse existing datasets with sufficient taxon sampling to detect Cnidaria+Placozoa is interesting and intuitive – it is somewhat of a mystery that this signal has not been previously reported in an area of intensive scrutiny (deep metazoan phylogenetics), if it is indeed legitimate. The point that there may be “taxon sampling and extensive differences in the datasets utilized used in previous analyses” is very important in this context. We could locate no published datasets that satisfied the following criteria, which would be needed for strict comparability to the present dataset:

1) Had adequate taxon sampling (10+ species) of all non-placozoan, pre-Bilaterian taxa and a good sampling of non-Metazoan outgroups (particularly Choanoflagellata).

2) Included Placozoa in orthology analysis.

3) Considered large numbers of orthologs (e.g. >2000 alignments), such as would result from global orthology analysis (typically, MCL^-^based approaches).

4) Made full, un-groomed (e.g. through alignment trimming) orthology groups publicly available.

One dataset which came close to satisfying these criteria was that of Simion et al., (2017). They do not, unfortunately, make their entire dataset available online as untrimmed orthologue groups (there would be 4,002 in this case), but they have made available a set of 1,719 trimmed alignments which display adequate taxon sampling to search for a signal of Placozoa+Cnidaria. On these alignments, we employed the p4 null simulation exactly as described in the Materials and methods section on our 1,388 genes. This yielded a set of 834 test-passing alignments, and 883 test-failing (with two genes that caused errors in the p4 simulation test, likely due to the inclusion of gap-rich sequences post alignment). Using MARE to make information rich submatrices, we end up with a 561-gene, 91,116 residue alignment in the test-passing set (indicating, again, a shorter average length in compositionally less biased genes), and a 382-gene, 119,650 residue alignment in the test-failing set. We started six chains under the CAT+GTR model in PhyloBayes MPI in both amino-acid and Dayhoff-6 recoded space in both matrices.

Unfortunately, we find that achieving convergence has been challenging in both of these matrices in the limited time we have had while preparing this revision (although we continue to run chains from both matrices). However, qualitatively, the independent chains within each dataset are largely identical in the relationships they show among the 5 major animal clades. The general conclusions appear to be:

1) In the test-failing matrices at the amino-acid level, Ctenophora are recovered with full posterior probability as the sister-group to the remaining animals, whereas under Dayhoff-recoding support for this position becomes low (pp 0.87 for instance) or, indeed, in some chains, Porifera are recovered with full support as this sister-group. In the test-passing matrices at the amino-acid level and the recoded levels alike, Porifera are consistently recovered with full posterior probability as the sister-group to the remaining Metazoa.

2) In no chain was Cnidaria+Placozoa recovered, in either amino-acid or Dayhoff-recoded space (instead the conventional Cnidaria+Bilateria relationship was seen).

A straightforward interpretation of this result would be that a.) as suggested later in the Feuda et al., (2017) manuscript, Ctenophora-sister is a compositionally-driven artefact and that b.) this published dataset does not contain any signal for a Cnidaria-Placozoa relationship. In our opinion, full confidence in the latter conclusion is difficult to muster, although these analyses are at least consistent with that interpretation. However, due to the substantial differences in methodology (orthology, alignment parameters, alignment trimming), and particular taxon sampling (e.g. this Simion et al., dataset lacking the additional clade A placozoan genomes but including many other divergent outgroups with distinctive compositional properties, which may have affected the results of the p4 simulation tests), it is difficult to completely compare these results to our own. A more consistent comparison would require the public availability of the full orthology dataset.

Nonetheless, this preliminary experiment does show that signal for the Placozoa–Cnidaria clade may indeed be variable among different taxon and orthologue sets. We have therefore chosen to somewhat temper the language in the manuscript, although in our opinion the consistent direction of the analyses we have shown of our own dataset is to favour this clade.

4) Results and Discussion section. The reported PPA scores are very bad, indicating that both datasets are extremely heterogeneous and that at the AA level both datasets are unreliable. The PPA do suggest BUSCO is worst, but that does not make your new AA dataset more reliable as Z-scores greater than one hundred indicate an utter failure of the model to describe the data (see Feuda et al., 2017 for a discussion of interpretation of Z-scores). This should be pointed out in your discussion. Note however, that the fact that a dataset is highly heterogeneous does not mean that the tree it supports is incorrect; that depends on the specific of the dataset (taxon specific distribution of heterogeneity, presence of other forms of noise etc.). High heterogeneity means that we should be cautious when interpreting the results as the tree might include tree-reconstruction artefacts. So, our reading of these two analyses is that both datasets are heterogeneous and that tests (like analyses under Dayhoff-6) need to be carried out to validate the clades in these trees. You did these tests and they clearly indicate that Placo-Cni and Pori-sister are unlikely to be compositional artefact while Cneto-sister is likely a compositional artefact.

We have directly acknowledged ("Both the initial 430-gene matrix and the 303-gene BUSCO matrix fail these tests") that both matrices suffer from compositional heterogeneity, judging from the posterior predictive tests. We also agree that while these scores indicate the BUSCO test fails to a greater degree, the fact that both matrices are so heterogeneous indicates that neither is strictly reliable – indeed, this was what motivated our per-gene simulation test. We agree with the reviewer’s assertion that the resilience of Cnidaria+Placozoa to Dayhoff-6 recoding indicates that this clade is less likely to be a compositional artefact.

5) Results and Discussion section: You need to provide more information about heterogeneity. Are you talking of site or lineage specific heterogeneity? These are two different components of a dataset’s heterogeneity that are differently modeled and both need to be considered. Site-specific heterogeneity is generally modelled relatively efficiently using CAT-GTR while lineage specific heterogeneity cannot be modeled using the standard models you used in this paper, as you would need BreakpointCAT, NDCH or similar to model it. D6 Recoding reduces both forms of heterogeneity and, in combination with CAT-GTR generally, allows for an adequate modeling of site-specific heterogeneity and an improvement in the modeling of lineage specific heterogeneity (the latter not necessarily reaching adequacy – see Feuda et al., 2017). Hence it is key that you report statistics for all your datasets (Z-scores for both site and lineage-specific heterogeneity for all you're a and Dayhoff datasets). A table with the Diversity and Comp PPA (these can be derived using Phylobayes) will do and should be easily done as you should already have all the chains (as you already run other PPA). A table presenting all z cores for all your dataset would be key to interpret your results (Figure 1 to Figure 4D included) and decide which tree is less likely to be incorrect/more likely to be accurate.

In context we believe it was clear that our usage of “heterogeneity” was referring specifically to time heterogeneity in among-lineage residue frequency, rather than site heterogeneity. We very much agree with the reviewer’s point that even the site-heterogeneous CAT+GTR model fails to adequately model compositional drift over time, and would further add that the other models cited, BP-CAT and NDCH, do not have sufficiently modern implementations (e.g. with MPI) to use on contemporary large-scale datasets, and/or have other major limitations (e.g. the NDCH implementation in p4 not being compatible with site-heterogeneous mixture modelling). Indeed, we used Dayhoff recoding specifically to help mitigate the influence of compositional bias in our CAT+GTR analyses – but we are in agreement with Feuda et al., that “mitigate” is indeed the proper term, since even Dayhoff recoding cannot fully remove all non-stationary signal in a dataset. We have added a citation in this paragraph to the Feuda et al., manuscript at the first mention of posterior predictive tests and also to a recent commentary piece published by the first author in Integrative and Comparative Biology, which contains an extended discussion of the potential value and limitations of recoding, which we hope may help further contextualize these results. We also agree that there is value in providing z-scores for diversity and composition posterior predictive analyses for the various matrices we have employed; for all 4 supermatrices discussed in the manuscript, we now summarize z-scores in Supplementary file 2. Unfortunately, such posterior predictive tests can only be presented for amino-acid level analyses – we ran the Dayhoff-recoded analyses in Phylobayes 4.1c (serial version), whose default run behaviour does not record the full chain file needed to undertake a posterior predictive test. Total reanalysis in D6-space would therefore be required to present these scores.

6) Results and Discussion section: This section reads like an ad hoc attempt to justify Ctenophora-sister, a result that is tangential to your paper (which is about Placozoa) and that is not supported by your analyses. Specifically:

We agree that the debate over Ctenophora-vs-Porifera sister group of all other metazoans is indeed tangential to this paper, and hope that this discussion does not derail the way this paper at large is evaluated. However, we would like to respond to this confrontation in clear terms: our specific claim is that the factors influencing whether Porifera vs. Ctenophora is recovered as the sister group to the remaining metazoan may vary among datasets, as it does in virtually all published data sets (including many that claim to resolve this). While in some supermatrices, we certainly do agree that compositional heterogeneity is likely to be the main driver, this cannot be the case for other datasets.

6A) Figure 3 presents the BUSCO trees under CAT-GTR+G (3A) is AA and (3B) is Dayhoff. Figure 3A (your worst overall dataset in terms of heterogeneity) clearly supports Ctenophora-Sister, while Figure 3B shows an unorthodox PORIFERA+CTENOPHORA (but with relatively low support ~ 75%). Neither of these two figures supports Porifera-sister. According to your arguments BUSCO is the worst dataset. When considered as an AA alignment, the BUSCO dataset maximizes its compositional heterogeneity and supports Cteno-sister. In the text you state that Figure 4A (the dataset of the heterogeneous genes in your dataset – that we will discuss further below) provides evidence that Pori-sister might be linked to heterogeneity. However, even if that was actually the case, then Figure 3A must be evidence that Cteno-sister is a compositional artefact (equally linked to heterogeneity). Yet this important fact is not even mentioned in the text. This is surprising and unfair given that you use the heterogeneity of the dataset used to derive Figure 4A to make an argument against Porifera-sister, the heterogeneity of Figure 3A to suggest Placozoa + Bilateria is likely to be an artefact, but you do not discuss the fact that also Cteno-sister emerges from the "bad" BUSCO dataset and should thus be considered to be dubious. The text needs to be modified to reflect this.

Firstly, we assume that “Placozoa + Bilateria” in this response was a typo, and that the reviewer meant to write “Placozoa + Cnidaria” – in no analysis did we recover such a clade. Secondly, we are in full agreement that the Ctenophora-sister result recovered in our BUSCO supermatrix is likely to be a compositional artefact *in this specific dataset*: both the posterior predictive analyses (which show higher z-scores for most ctenophores as well as some calcisponges in the BUSCO dataset) and the fact that support for Ctenophora-sister erodes under Dayhoff recoding are highly suggestive on this conclusion, as the reviewer states.

6B) The experiment in Figure 4 is nicely designed, but some aspects of its implementation and the interpretation of its results are problematic. Specifically, we think that the analyses reported in Figure 4A and 4C are misleading for two reasons. First, the data matrix cannot be proved to be composed only by heterogeneous genes. This is because 111 of the 764 genes in the dataset used to derive this figure (~ 14% of the superalignment) are composed of genes that are not, strictly speaking (i.e., at the 0.05 cutoff level), heterogeneous. Second, even if this dataset was composed purely of heterogeneous genes it could not be used to claim, for example, that Porifera-sister is linked to heterogeneity. This is because on this dataset Dayhoff recoding did not change the position of the sponges. This indicates that the signal for this node in this dataset (irrespective of its heterogeneity) is not driven by the within-D6classes changes that are known to be associated with compositional heterogeneity. Irrespective of the heterogeneity of the dataset, the fact that Porifera sister is unchanged and does not loose support upon recoding indicates that it is supported by between D6-classes changes, which are not silenced by recoding, and are more likely to be associated phylogenetic signal. To conclude, you do not have evidence in Figure 4A and 4C to claim that the assertion of Feuda (that Cteno-sister is likely to be associated with compositional heterogeneity while Porifera-sister is not likely to be a compositional artifact) might be incorrect. The Dayhoff test rejects your conclusion. So, what to do with the analysis of Figure 4A and 4C? Our thought is that this dataset is confusing your experiments because it is composed of a mixture of more and less heterogeneous genes and it is best excluded. We suggest that you just report the results in Figure 4B and 4D which are actually very clear and say all that there is to say on Placo-Cni and Pori-sister vs. Cteno-sister.

Firstly, we must emphasize that we never have claimed that this experiment tells us that Porifera-sister is likely to be a compositional artefact (“that Porifera-sister is linked to heterogeneity”, in the reviewer’s words). Our specific claim was only that in the context of this experiment, we found it difficult to understand Ctenophora-sister as the result of compositional non-stationarity.

Indeed, we would agree that since Dayhoff recoding seems to diminish support for Ctenophora-sister in the test-passing matrix (Figure 4D), while Porifera-sister remains unchanged by the recoding in the test-failing matrix (Figure 4C), probably Ctenophora-sister is more likely to be the artefact. Instead what we are saying is that our results question the assertion of Feuda et al., 2017 that the converse is true in general: we see no evidence that Ctenophora-sister can be interpreted to be the result of compositional heterogeneity in this dataset (Figure 4). If this were true, we'd expect to see Ctenophora-sister in Figure 4A even in a CAT+GTR analysis, because CAT+GTR is also a model that assumes compositional stationarity. We find it very helpful to consider that we actually do recover strong support for Ctenophora-sister in an amino-acid analysis of our test-failing 349-gene matrix: in our profile mixture model ML tree made from this dataset (Figure 4—figure supplement 1). It is only when we analyse this same matrix under CAT+GTR that we recover Porifera-sister. Because both the ML model and CAT+GTR make the assumption of stationarity, logically, it can’t be that Ctenophora-sister is recovered under ML alone because of compositional heterogeneity. Therefore, in this matrix, it’s more likely that this result is driven by some other artefact – for instance, saturation due to site-heterogeneous substitution, which is better modelled by CAT+GTR than the profile mixture model.

Moreover, while in general we agree that Dayhoff-6 (and other forms of) recoding will mitigate compositional heterogeneity, we would also like to remark that a.) some compositionally-driven changes will still be apparent in between-Dayhoff-group substitutions and that b.) this may not be the only form of systematic error against which recoding might be protective. It might also well be that the recoding-driven erosion of support we see for Ctenophora-sister comparing e.g. Figure 4A and 4C (and elsewhere) is actually due (in some part) to a different class of systematic errors, e.g. heteropecilly or heterotachy, which also happens to be diminished under Dayhoff recoding. This is discussed at greater length in the recent ICB article by the first author. The statistical properties of recoded datasets are not yet completely understood, and we simply remark that it is a bit long-reaching to assert that the sole effect of recoding is to control compositional bias.

The other comment the reviewer makes here, “that the analyses […] are misleading” because “the data matrix cannot be proved to be composed only by heterogeneous genes” is difficult to respond to. No statistical test guarantees completely accurate results, and it is true that due to some type I error (inappropriate rejection of the null hypothesis), some proportion of genes in the test-failing matrix actually fit the stationarity assumption well. It is also true that some heterogeneous genes may have been included in the test-passing set due to type II error (although we have tried to minimize this by selecting an α-threshold of 0.10).

We believe that readers with a background in basic statistics will be able to critically assess the results displayed in Figure 4 themselves, and respectfully disagree with the reviewer’s suggestion that figures 4A and 4C are “best excluded” – it is only in comparison with figures 4B and 4D that this experiment shows the Cnidaria–Placozoa clade to be associated with compositional heterogeneity, which is the central point of this paper.

6C) The fact that the genes in your "homogeneous dataset" are individually homogeneous does not mean that your homogenous superalignment is in itself homogeneous. This needs to be tested (using PPA under CAT-GTR for both Dayhoff-6 and AA datasets). This is because heterogeneity in a multigene dataset adds up and it is customary that superalignments composed of individually homogeneous genes fail heterogeneity tests. My expectation is that the dataset in Figure 4B will be shown to be still heterogeneous and that of Figure 4D will be shown to be much more homogeneous (as it is always the case with D6-recoded datasets). Accordingly, the analyses of Figure 4BD indicate that there is evidence for Cteno-sister to be a compositional artefact (driven by within D6-classes changes only present in Figure 4B) while there is no evidence for Cnidaria+Placoza and Pori-sister to be compositional artifacts (as they are both present in Figure 4D where only between classes changes are evident). In addition, this analysis confirms the existence of two signals in the data with reference to Ctenophores (one linked to unreliable within D6 classes changes) and one linked to the more reliable between classes changes, while Placo-Cnidaria seem to be the only signal in the dataset and it is more strongly represented in more reliable between-classes changes.

We certainly do agree with the reviewer’s prediction that a concatenated matrix composed of genes that individually pass a test of compositional heterogeneity might still, in aggregate, show evidence of compositional nonstationarity. Indeed, this is what the posterior-predictive analysis presented in the new Table 2 demonstrates: all amino-acid super-matrices presented in this paper (and in others, see e.g. Feuda et al., 2017) fail to meet the stationarity assumption of the CAT+GTR model. However, as predicted, the matrix from the test-passing set has the lowest mean z-score. Presumably, our recoded analyses in all matrices will have further mitigated (but not eliminated) such non-stationary compositional variation; however, as described above, we are unable to report the results of posterior predictive tests with under Dayhoff recoding without fully redoing these analyses.

7) Results and Discussion section: Here you seem to assume that Dayhoff recoding always have to change the topology and that if the topology does not change the recoding, in some way, it failed. The application of D6 was "not significant" (in your words) for the dataset of Figure 4AC. The problem with your statement is that a tree does not need to be incorrect simply because the data are heterogeneous, hence D6 does not need to invariably cause topological changes. So, its application was not "not significant", but it did not drive to a topological change. As said above as well, D6 reduces the influence of compositional heterogeneity by silencing within category changes. The analysis in Figure 4AC indicates that Porifera-sister is driven by more reliable among category changes which are generally not compositionally driven. Otherwise it would have disappeared in the Dayhoff analysis. Note that we are not saying that Pori-sister is correct; rather, we are saying that Figure 4A C does not provide evidence that this clade is compositionally driven, hence you cannot conclude that Porifera-sister might be associated with compositional attractions. Note that, with reference to the Ctenophora debate, the application of recodings has invariably produced strongly directional changes Ctenophora->Porifera. This is a highly non-random directional change and at this time there is no known Dataset that while supporting Porifera-sister at the AA level (under CAT-GTR) was found to support Cteno-sister after recoding. This is true of all the datasets of Whelan when Choanoflagellata are the only outgroups used (pori-sister as AA and as Dayhoff), and for your dataset in Figure 4AC – for example. This can only be interpreted as suggesting that in all available datasets Porifera-sister is never driven by within category changes, accordingly, when it emerges at the root, is cannot be because of a compositional attraction – irrespective of the heterogeneity of the dataset itself. Differently, D6 experiments invariably indicate that when ctenophores emerges at the base, compositional heterogeneity can never (all datasets tested to date behave exactly in this same way -including yours) be ruled out – strongly suggesting Cteno-sister to be a compositional artifact.

This reviewer again reads too much into the text when he remarks that “you seem to assume that Dayhoff recoding always have to change the topology and that if the topology does not change the recoding, in some way, it failed”. Our usage of the phrase “not significant” in the context of Figure 4A and 4C was simply descriptive: we were only remarking that the topologies and supports do not change due to D6 recoding. Indeed, we agree that this probably indicates that Porifera-sister, in this dataset, is the result least likely to be the result of a systematic error, precisely because it is robust to Dayhoff recoding.

As an aside, although we do not see the relevance *per se* to this manuscript, the reviewer’s comment that “this is a highly non-random directional change and at this time there is no known Dataset that while supporting Porifera-sister at the AA level (under CAT-GTR) was found to support Cteno-sister after recoding” is not entirely true. In unpublished data, we have seen one large supermatrix which has poor (0.87 pp) support for Ctenophora-sister under amino-acid CAT+GTR analysis, which actually increases to full posterior probability under CAT+GTR of the Dayhoff-recoded version of this same matrix. This shows that the “directionality” of results from recoding is not necessarily consistent among datasets, and in our minds highlights that the theoretical properties of recoded alphabets are still not yet completely understood. Interestingly, however, in a separate analysis, we also recover Cnidaria+Placozoa in one analysis (using only the H1 reference strain as a representative) – in the matrix in question, this clade is recovered only when all non-choanoflagellate outgroups are removed and when the data are recoded into D6 categories – an experiment which was actually inspired by one of the reviewers’ past papers (Pisani et al., 2015), and which also suggests using an independent dataset that Cnidaria+Bilateria may be a compositionally driven artefact.

8) Title: Given how controversial the relationships at the base of the animal tree have been, perhaps a more moderate title would be more fitting (e.g., Phylogenomic support for a sister relationship of Placozoa and Cnidaria).

The suggestion is a good one; we use a more nuanced title in the resubmission.

9) The scripts used for data processing in genome assembly, phylogenomics matrix construction and analysis are not linked in the paper. Sharing these scripts on a publicly available repository like Github would enhance the reproducibility of the present study.

All python scripts used in producing and curating these datasets were actually included in this submission with the supplementary data available in the DataDryad accessory, as stated in the “Source Data Availability” section. Indeed, as these were for the most part one-off helper scripts (sometimes hard-coded), they are most interpretable/re-usable when bundled alongside the data they were used to manipulate. We understand that this DataDryad accession was available to reviewers and should have been available for inspection during the review process. If editorially required, we are certainly happy to also make these scripts available in a github page or elsewhere.

[Editors' note: further revisions were requested prior to acceptance, as described below.]

Essential revisions:1) The authors allay some concerns over the description of the genome data and the point that the new genome assemblies produce BUSCO orthologue occupancies that are similar to other transcriptome assemblies is appreciated. One does wonder why short read genome assemblies, which produced highly fragmented assemblies, were utilized in the first place, instead of transcriptomes. If the purpose of doing so was to enhance placozoan genome resources, these resources are not described here in any detail because they are too fragmented. "we opted not [sic] to omit a detailed characterisation of these genomes, partially out of spatial constraints (this article was submitted as a Short Report format), and partially since the genomes themselves are Illumina short-insert-only assemblies and therefore relatively fragmented". Please provide this information.

Our purpose was explicitly not to provide new placozoan genomic resources in this paper – indeed, it seems this has been already reasonably adequately done during this article's review process (Eitel et al., 2018), although the authors of that paper regrettably did not take the opportunity to use their data to independently test our Cnidaria+Placozoa hypothesis which had been expressed in preprint form and was known to them. In reality, the sequencing data from these short-insert libraries were originally generated by Gruber-Vodicka and colleagues as part of another ongoing project. However, as is standard in the field, genomic data are very re-usable, and because placozoans seem to have relatively simple repeat landscapes and short introns, we saw that we could get acceptable enough gene representation from these fragmented assemblies to perform a phylogenetic analysis (Table 1). Indeed, the high occupancy of all placozoans we sample in our matrices (see bar plot on right of Figure 1) seems to have validated this intuition. We have altered the text to make the provenance of these data more transparent to the reader.

2) The manuscript now includes information of the% age, means and standard deviations for partition lengths. This information is important for evaluating the possible causes of the phylogenetic discrepancies (e.g. Cnidaria+Placozoa vs. Cnidaria+Bilateria) observed in different datasets. We now see that dataset that supports Cnidaria+Placozoa are systematically composed of shorter partitions. Thus, compositional bias is not the only difference between the two datasets. No further explanation is given. Please report these data and explicitly discuss the finding that ompositional bias is not the only difference between the two datasets.

We agree with the reviewers' assertion that it would benefit the transparency of this paper to emphasize this observation more explicitly, and in this resubmission we have brought attention to this in the text. However, "no further explanation is given" because, in truth, how to explain this correlation remains unclear to us. It is, however, worthwhile in assessing these results to consider recent literature seeking to understand the influence of intrinsic gene properties on phylogenetic inference. There appear to be relatively few papers that systematically assess correlations between different sequence-based metrics and their influence on phylogenetic inference, but we were particularly struck by one recent paper for which the reviewing editor was the senior author, Xing-Xing et al., 2016. This paper attempted to systematically investigate the correlation structure and relative importance to phylogenetic inference of 52 different quantitative properties of multiple sequence alignments. Among other results, they found that only two sequence-based properties, alignment length (with or without gaps), and Relative Composition Variability (RCV), a measure of compositional homogeneity, were strong predictors of phylogenetic signal and consistency with assumed “true” trees (eMRC trees in yeast and mammal datasets). These authors also found, evident in their Supplemental Tables S4 and S5, that these two metrics are strongly (-0.4388518 for yeast, and -0.1354731 for mammals) negatively correlated. A reader might therefore conclude that longer genes are more phylogenetically informative, less likely to be compositionally biased, and more consistent with the true tree – and in light of this conclusion, might reasonably question the Cnidaria+Placozoa result found in our supermatrices composed of systematically shorter genes (Figure 1 and Figure 2, Figure 4B and 4D).

To make a fair comparison against the results of Xing-Xing et al. 2016 and these unpublished results, we also plotted the RCFV score, a refinement of the RCV metric designed to accommodate gapped alignments, introduced by Zhong et al., 2011, against gene length in our 1,388 orthologue set.

Intriguingly, using RCFV as a quantitative index of compositional heterogeneity, we see no evidence of a trend against gene length (from a linear regression, R^2^ = 0.000128562, adjusted R^2^ = -0.000592846), running against both of the above-cited results. Nonetheless, our previous report that it is, in general, smaller genes that pass the p4 compositional heterogeneity null simulation test (with p > 0.10), is clearly evident in the above plot, despite similar overall RCFV scores to the test-failing population.

Our conclusion at the moment, considering trends within our own data and the published literature, is that no general statements can yet be made regarding the relationship between alignment length and the degree of compositional bias. As the above plot indicates, even within one dataset results may differ depending on which metric of non-stationarity is used (although we expect that the p4 test, as it explicity attempts to incorporate gene tree topology, is the more sensitive than a simplistic, expectation-agnostic RCFV distribution thresholding). Furthermore, it is unclear to us that there exists a general relationship, whether constant or variable with scale of evolutionary divergence, between compositional bias and gene length. This is clearly an area in need of further investigation with empirical and theoretical approaches, hopefully incorporating multiple metrics of bias. It is our hope that the publication of this work, which cannot have the ambition of being a systematic phylogenetic methods comparison, might nonetheless motivate further such systematic studies. At present, we see no particular reason from first principles or a consensus of empirical studies to devalue results from supermatrices composed of shorter partitions.

3) The paragraph describing metaBAT and metawatt does not explicitly mention contamination, which would help the reader interpret the purpose of these steps. Please rectify this.

We agree this would help clarify the purpose of these procedures in the context of this phylogenetic paper (a discipline where metagenomics tools are not often used). We have altered the text accordingly.

4) We agree that Dayhoff-6 recoding is not a silver bullet for compositional bias and that previous analyses that have not filtered data for compositionally unbiased partitions are not direct comparisons. However, the re-analysis of the Simeon dataset that the authors conducted, where unbiased partitions were selected and analyzed in both amino acid and Dayhoff-6 recoded space, does represent a more or less direct comparison and these results strongly contradict the topology favored by Laumer et al. While there are expectedly differences in taxon sampling and data handling between the Simeon re-analysis and that presented by Laumer et al., but this is true of the vast majority of previous phylogenomic analyses at this scale which confidently resolve other nodes in the animal tree of life. The explanation for this in the author's response, "this published dataset does not contain any signal for a Cnidaria-Placozoa relationship" seems quite correct indeed, but phylogenomic datasets as large as Simeon (which is much larger than that presented by Laumer et al.,) should possess such phylogenetic signal if it were robust. Therefore, this re-analysis of the Simeon dataset, which is not included in the revision, draws the central phylogenetic finding of Laumer et al., into question and strongly suggests that some other feature of the Laumer et al., dataset is contributing to Cnidaria+Placozoa. As stated above, the unbiased partitions are significantly shorter than the biased partitions. It is possible that this finding is somehow, perhaps indirectly, related to the central finding?

As we attempted also to convey in our previous response, we disagree that our reanalysis of the Simion et al., dataset "does represent a more or less direct comparison" to our own analyses. Differences in taxon sampling and data handling matter, and there are multiple plausible explanations for why we failed to find support for Placozoa+Cnidaria in our reanalysis. Those at the top of the list for us include:

a) The Simion et al., genes included more distant, non-choanoflagellate outgroups, which include exceptionally compositionally biased taxa (e.g. Filasterea such as *Capsaspora*; see z-scores from PPA in Nosenko et al., 2013). This might influence not only phylogenetic inference (in particular by skewing the mean amino acid frequency vector) but also the p4 composition test.

b) The Simion et al., genes were provided to the public pre-aligned and pre-trimmed (https://github.com/psimion/SuppData_Metazoa_2017). The effect of multiple alignment algorithms (and alignment masking) on phylogenetic inference was extensively explored, and shown to be highly significant in driving results in the pre-genomic era, but it has largely been ignored by authors working with large-scale datasets. Of particular note, Simion et al., used a MAFFT algorithm different from ours (L-INS-i vs. our E-INS-i), and masked their alignments with BMGE, presumably using default parameters (which notably do not include its option to trim compositionally biased sites), representing a more stringent alignment masking (as it incorporates a measure of “entropy” in its trimming) than our use of ZORRO.

c.) The Simion et al., dataset, naturally, did not include any clade-A placozoan representatives. While we give one anecdotal analysis (Figure 1—figure supplement 2) suggesting that increased taxon sampling of Placozoa was not driving the Placozoa+Cnidaria clade in our dataset, it is not yet clear to us that inclusion of the total extant diversity of Placozoa has no influence on phylogenetic inference (especially in light of compositional bias) generally. We would be particularly interested in how taxon sampling (particularly of compositionally deviant taxa such as non-choanoflagellate outgroups and within Placozoa) might influence the outcome of a compositional bias test.

For these reasons, and additionally for the important reason that these reanalyses did not achieve convergence using the CAT+GTR model in a reasonable time (note that the original Simion et al., paper reports only results from the simpler and computationally more tractable CAT model, not CAT+GTR), we chose not to explicitly report them in our revised version of this Short Report, as we did not wish to invite direct comparisons between these results and our own.

5) Again, while Ctenophora-vs.-Porifera is tangential to the central claims of the paper, I did find it relevant that in the authors’ response (unpublished data), Cnidaria+Placozoa was recovered in "one" analysis, but only when certain outgroups were removed, and the data were D6 recoded. This finding, like the Simeon reanalysis, speaks strongly to the lability of Cnidaria+Placozoa. Please comment on this issue.

We are somewhat hesitant to comment more exhaustively on this, since in our opinion during the review process an article should be evaluated on its own merits, not in relationship to other work, however related, being considered for publication elsewhere. However, we brought it up explicitly since it seemed a straightforward, evidence driven counter-argument to the review of Pisani.

What we omitted to do in summarily describing our unpublished data in this forum, was to mention its second CAT+GTR analysis showing strong support for Cnidaria+Placozoa, displayed in a Supplemental Figure. The analysis in question – based on an unrecoded amino acid matrix, in contrast to the other experiment we described in our initial review response – comes from a 43K submatrix of an originally ~100K residue supermatrix, created by leaving non-choanoflagellate outgroups (plus a few phylogenetically redundant and data-poor metazoan taxa) in the matrix prior to BMGE trimming of saturated and non-stationary sites. In contrast, deleting these outgroups from the alignment and then applying the BMGE algorithm yields a ~56K site matrix, which shows strong support for Cnidaria+Bilateria.

The reviewers are quite correct to point out "the lability of Cnidaria+Placozoa", a fact which we tried to emphasize in this manuscript as written, and which is also apparent within the analyses of our unpublished data. What is important to us is to consider the directionality of the lability: Placozoa+Cnidaria does not appear randomly as one compares different analyses, but instead shows up only in those analyses which were constructed specifically to minimize the influence of compositional bias: either by testing and excluding biased genes individually, trimming biased individual sites from a concatenated matrix, removing compositionally biased distant outgroups whose inclusion might skew the inferred residue frequency vector, recoding data into Dayhoff groups, or some combination of these. In the absence of phylogenetic inference software which is able to efficiently handle both site-heterogeneous and time-heterogeneous substitution, such matrix curation choices are our best current defence against systematic error. The fact that Placozoa+Cnidaria – a clade which, we emphasize, has not even been reported in recent molecular phylogenetic literature – shows up only when such stringent measures are taken is indeed at the heart of our argument that this clade is the more credible of the two we have observed, despite its lability to analytical conditions. We consider that this information will be extremely helpful for evaluating future phylogenetic hypotheses, not just in relation to Placozoa and Cnidaria.